# Autonomous Evaluation of LLMs for Truth Maintenance and Reasoning Tasks

**Rushang Karia**[∗]**, Daniel Bramblett,**[∗] **Daksh Dobhal, Siddharth Srivastava**
School of Computing and Augmented Intelligence
Arizona State University
`{rushang.karia,drbrambl,ddobhal,siddharths}@asu.edu`

## ABSTRACT

This paper presents AutoEval, a novel benchmark for scaling Large Language Model (LLM) assessment in formal tasks with clear notions of correctness, such as truth maintenance in translation and logical reasoning. AutoEval is the first benchmarking paradigm that offers several key advantages necessary for scaling objective evaluation of LLMs without human labeling: (a) ability to evaluate LLMs of increasing sophistication by auto-generating tasks at different levels of difficulty; (b) auto-generation of ground truth that eliminates dependence on expensive and time-consuming human annotation; (c) the use of automatically generated, randomized datasets that mitigate the ability of successive LLMs to overfit to static datasets used in many contemporary benchmarks. Empirical analysis shows that an LLM's performance on AutoEval is highly indicative of its performance on a diverse array of other benchmarks focusing on translation and reasoning tasks, making it a valuable autonomous evaluation paradigm in settings where hand-curated datasets can be hard to obtain and/or update.

## 1 INTRODUCTION

Large Language Models (LLMs) have been demonstrated to perform well in many natural language tasks involving formal languages such as *autoformalization* – converting natural language (*NL*) to formal language (*FL*) such as source code, math etc., (Wu et al., 2022; Liang et al., 2023), *informalization* – converting *FL* to *NL* (e.g. code summarization), and *reasoning* – using LLMs to perform sound reasoning or derive proofs. Although these methods have been successful in small-scale scenarios, LLM's effectiveness in maintaining factual accuracy or preserving which facts are true across translation remains unclear due to the difficulty in designing benchmarks that capture truth maintenance in such tasks. Multiple authors have noted that existing benchmarks and evaluation methodologies for such tasks are susceptible to the Benchmark Contamination Problem due to their use of static datasets, e.g., HumanEval (Chen et al., 2021; Wu et al., 2022; Han et al., 2024), and/or metrics that are insufficient/incomplete syntactic measures of evaluation (e.g, BLEU scores (Callison-Burch et al., 2006). As a result, existing methods provide misleading signals on the capabilities of LLM technology. One effective method to mitigate this problem in existing benchmarks is creating new data (Xu et al., 2024a). This is a tedious and expensive process since data generation requires expert annotators to hand-generate well-balanced datasets. While using LLMs as judges and/or metrics is a promising research direction (Zheng et al., 2023; Shankar et al., 2024; Xu et al., 2024b; Madaan et al., 2023), it is unknown whether LLMs can be used as accurate verifiers.

This paper addresses three key desiderata for benchmarking LLM capabilities in truth maintenance across *NL* and *FL*: *(D1) Can we dynamically generate out-of-distribution datasets without human annotators? (D2) How do we accurately assess an LLM's truth maintenance capabilities? (D3) Can we develop a benchmark predictive of LLM performance on formal translation and reasoning tasks?*

**Main contributions** Our key contributions are as follows:

1. A new approach for automatic synthesis of well-balanced test datasets using context-free grammars that are unlikely to be memorized during the LLM's training process (§D1).

---

[∗]These authors contributed equally.

2. The utilization of formal verifiers such as theorem provers to provably validate syntax-independent notions of correctness without having to exhaustively test over all possible truth valuations of formal syntax involving logic (§D2).

3. ∀uto∃∨∧L: a scalable, plug-and-play assessment system for benchmarking new LLMs as and when they are developed. Our system can be extended to any class of formal syntax that uses a grammar and admits an equivalence checker.

4. We show that LLM performance on our metric serves as an effective indicator of LLM performance on other metrics across a wide variety of tasks, such as first-order logic reasoning (§D3). Thus, our metric offers a scalable and efficient surrogate for evaluating new LLMs in tasks where other metrics may be limited due to the unavailability of new datasets. We also show that SOTA LLMs are unable to maintain truth effectively.

## 2 FORMAL FRAMEWORK

Large Language Models (LLMs) are non-linear functions represented by (billons of) parameters $\theta$ that, given a set of input tokens $x_1, \ldots, x_n$, typically from natural language *NL*, predict the output token $y_{i+1}$ using the distribution $P(y_{i+1}|x_1, \ldots, x_n, y_1, \ldots, y_i; \theta)$. The input tokens contain *context* $\kappa$ (also known as a prompt) that provides the necessary information for the task (e.g., instructions, etc). It is known that $\kappa$ significantly impacts the response quality $y_1, \ldots, y_n$ (Sahoo et al., 2024).

Propositional Logic is a branch of logic that utilizes *propositions* and *logical operators* (e.g., conjunction: $\wedge$, etc) to construct sentences that can be used to perform reasoning using the rules of logic. For example, propositions, $p_1 = $ *It is raining*, $p_2 = $ *It is sunny* can be used to create a sentence $P = p_1 \vee p_2$. If $P$ is true and $\neg p1$ is observed, then one can use the rules of inference to deduce that $p_2$ is true (Huth & Ryan, 2004). Two sentences in propositional logic, $P_1$ and $P_2$, are equivalent, $P_1 \equiv P_2$, iff their truth values agree for all possible assignments. E.g., $\neg(p_1 \wedge p_2) \equiv \neg p_1 \vee \neg p_2$ since $\forall p_1, p_2 \in \{True, False\} \times \{True, False\}$, $\neg(p_1 \wedge p_2) = \neg p_1 \vee \neg p_2$.

First-order Logic (FOL) differs from propositional logic in that sentences are constructed using *predicates*, *quantifiers*, *constants*, *symbols*, and *variables*. A popular example is the syllogism, where, given two FOL sentences $\forall x. \, Man(x) \rightarrow Mortal(x)$ and $Man(Socrates)$, one can conclude that $Mortal(Socrates)$. A FOL sentence $F$ can be interpreted using a universe $\mathcal{U}$, a substitution operator $\sigma$, and an interpretation function $\mathcal{I}$ (Russell & Norvig, 2020). Two FOL sentences, $F_1, F_2$, are equivalent, $F_1 \equiv F_2$, iff they are equivalent under all possible models. E.g., $\neg \forall x. \, Man(x) \equiv \exists y. \, \neg Man(y)$.

A regular expression (regex) is a sequence of characters used to determine whether a particular string matches the pattern or *language* induced by the regex. For example, the regex $200(00)^*1$ using $\Sigma = \{0, 1, 2\}$ matches all strings possible using $\Sigma$ that begin with a two, followed by one or more pairs of zeroes, and end with a one (Hopcroft et al., 2001). Two regexes, $R_1$ and $R_2$ are equivalent, $R_1 \equiv R_2$, if they represent the same language. It is known that $R_1 \equiv R_2$ if their corresponding minimal deterministic finite automata, $D_1, D_2$, are isomorphic, i.e., $D_1 \simeq D_2$ (Hopcroft et al., 2001).

We refer to sentences (strings) in first-order and propositional logic (regexes) as formal language *FL* in this paper. We now provide a definition of *(Auto/In)formalization* in the context of LLMs.

**Definition 2.1** (Autoformalization: $\mathcal{A}$). Given an LLM $L$, a *NL* $N$, a *FL* $F$, a string $\psi \in NL$, and context $\kappa'$, autoformalization $\mathcal{A}$, is defined as using $L$ to translate $\psi$ to $\varphi = \mathcal{A}_L(\psi, \kappa')$ s.t. $\varphi \in FL$.

**Definition 2.2** (Informalization: $\mathcal{I}$). Given an LLM $L$, a *NL* $N$, a *FL* $F$, a string $\varphi \in FL$, and context $\kappa$, informalization $\mathcal{I}$, is defined as using $L$ to translate $\varphi$ to $\psi = \mathcal{I}_L(\varphi, \kappa)$ s.t. $\psi \in NL$.

*Example* One possible autoformalization of *"Every human drinks coffee but some are not dependent on it"* in FOL is $[\forall x. \, Human(x) \implies Drinks(x, Coffee)] \wedge [\exists y. \, Human(y) \wedge \neg Dependent(y, Coffee)]$. Ideally, informalization will be an inverse of autoformalization. Therefore, the FOL formula $[\forall x. \, Human(x) \implies Drinks(x, Coffee)] \wedge [\exists y. \, Human(y) \wedge \neg Dependent(y, Coffee)]$ can be informalized to the sentence *"Every human drinks coffee but some are not dependent on it"*.

We assume that the context $\kappa, \kappa'$ provided contains the prompt and any necessary vocabulary that is needed for the task (e.g., *Human*$(x)$ represents that $x$ is a human, etc.). We omit $\kappa, \kappa'$, and $L$ in the notation for $\mathcal{A}$ and $\mathcal{I}$ where they are clear from the context.

Informalization and autoformalization are non-deterministic functions. Therefore, it is possible that a different LLM (or the same LLM with a different seed) autoformalizes the same input text to a syntactically or even semantically different output. E.g., the example above could be autoformalized to the semantically equivalent form: $\forall x.\ Human(x) \implies Drinks(x, Coffee) \wedge \neg \forall y.\ Human(y) \implies Dependent(y, Coffee)$. Similarly, an LLM can informalize differently. The example above could be informalized by the same LLM to *"All humans drink coffee but some are not dependent on it"*. Thus, the informalization (autoformalization) of an autoformalization (informalization) of a string is possibly different from that string: $\mathcal{A}_L(\mathcal{I}_L(\varphi, \kappa'), \kappa) \neq \varphi$ and $\mathcal{I}_L(\mathcal{A}_L(\psi, \kappa), \kappa') \neq \psi$. Given $n \in \mathbb{N}^+$, let $(\mathcal{A} \circ \mathcal{I})^n(\varphi_0)$ to refer to the sequence $\varphi_0 \to \psi_0 \to \ldots \to \varphi_n$ that is obtained using an LLM $L$ when starting with *FL* $\varphi_0$, where $\psi_i = \mathcal{I}(\varphi_i)$ and $\varphi_{i+1} = \mathcal{A}(\psi_i)$.

While syntactic differences across $(\mathcal{A} \circ \mathcal{I})^n(\varphi_0)$ operations may be acceptable, the ability of an LLM to maintain semantic content across $(\mathcal{A} \circ \mathcal{I})^n(\varphi_0)$ for *FL* such as first-order logic, regular expressions, etc., is foundational and underlies many aspects of the capabilities of LLMs surrounding reasoning, semantically accurate translation, etc. For programming, it has been shown that autoformalization accuracy is indicative of the reasoning abilities of LLMs since they frame reasoning as generation of *FL* (Chen et al., 2021). Others (Wu et al., 2022) have made similar observations and have highlighted the need for benchmarks and metrics for assessing the truth maintenance capabilities of LLMs. In this paper, we further show through our empirical evaluation that an LLM's ability to preserve factual information or semantic truth across translations is indicative of its performance on related tasks.

Intuitively, truth maintenance captures an LLM's ability to preserve truth across translation; operationally, it evaluates the ability of a system to be able to accurately invert its own translations. We say that an LLM maintains truth in translation iff $(\mathcal{A} \circ \mathcal{I})^n(\varphi_0)$ always leads to a $\varphi_n$ that is semantically equivalent to $\varphi_0$. Recall that $\equiv$ denotes the semantic equivalence operator in FL. Formally,

**Definition 2.3** (LLM Truth Maintenance). An LLM $L$ maintains truth in translation iff $\forall \varphi_0, n$, and for all sequences $(\mathcal{A} \circ \mathcal{I})^n(\varphi_0)$ obtained using $L$, $\varphi_n \equiv \varphi_0$.

In practice, we estimate the ability for truth maintenance through a sampling-based process. Naturally, LLMs may not autoformalize, reason, etc., correctly due to issues like hallucination (Ji et al., 2023), etc. For the earlier example, the LLM could autoformalize by omitting the $Human(y)$ statement to yield $[\forall x.\ Human(x) \implies Drinks(x, Coffee)] \wedge [\exists y.\ \neg Dependent(y, Coffee)]$. This seems innocuous but changes the meaning since $y$ is no longer required to be human, and thus it interprets as *"All humans drink coffee, but some element of the universe is not dependent on coffee."* Such issues have profound implications in synthesizing specifications and/or programs. Thus, an LLM must be able to understand its own generated output across *NL* and *FL*, and it is imperative to create a benchmark that can faithfully assess the truth maintenance of LLMs.

## 3 THE ∀UTO∃∨∧L APPROACH FOR ASSESSING TRUTH MAINTENANCE

We now describe our approach, ∀uto∃∨∧L, for autonomously assessing an LLM's ability for truth maintenance. ∀uto∃∨∧L provides dynamically generated datasets that can be scaled arbitrarily by systematically generating out-of-distribution, well-balanced ground-truth data (§D1 – Sec. 1), provides §D2 by using intrinsic LLM capabilities to automatically assess $(\mathcal{A} \circ \mathcal{I})^n(\varphi_0)$ without requiring any labeled annotations and using formal verifiers to rigorously check and guarantee the correctness of $(\mathcal{A} \circ \mathcal{I})^n(\varphi_0)$ without having to engage in an exhaustive search process.

### 3.1 AUTOMATIC EVALUATION OF TRUTH MAINTENANCE

We develop a novel technique that can soundly assess truth maintenance without any human annotations by evaluating $\varphi_i \to \psi_i \to \varphi_{i+1}$. Our approach is based on the following intuition. Let $\mathcal{I} : \varphi \to \psi$ be a non-deterministic function that maps *FL* $\varphi$ to *NL* $\psi$. Similarly, let $\mathcal{A} : \psi \to \varphi$ be a non-deterministic function that maps *NL* $\psi$ to *FL* $\varphi$. In general, there are many possible correct informalizations (autoformalizations) of $\varphi \in FL$ ($\psi \in NL$). Because $\mathcal{I}$ and $\mathcal{A}$ are non-deterministic functions, their inverses are thus not well-defined.

Our key observation is that if $\mathcal{I}$ and $\mathcal{A}$ come from the same system (e.g., an LLM), then we can evaluate that system's truth maintenance by *composing* $\mathcal{I}$ and $\mathcal{A}$. Let $\varphi$ be any *FL* expression and let $L$ be an LLM. If $L$ preserves truth, then $\psi = \mathcal{I}(\varphi)$ will be an accurate *NL* representation of $\varphi$ and

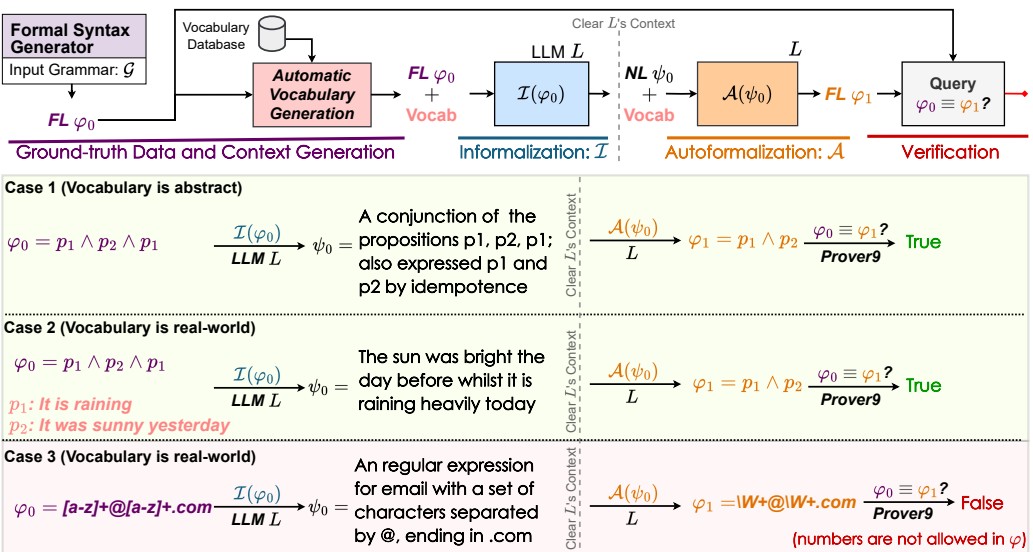

Figure 1: The ∀uto∃∨∧L process for autonomous evaluation of LLM truth maintenance w.r.t. $(\mathcal{A} \circ \mathcal{I})^n(\varphi_0)$.

$\varphi' = \mathcal{A}(\psi)$ will be a semantically equivalent *FL* representation of $\psi$. Since $\psi$ is an *NL* description, it is quite challenging to check whether $\mathcal{I}(\varphi)$ is indeed an accurate representation of $\varphi$ without human intervention. However, if $L$ preserves truth, $\varphi' = \mathcal{A}(\mathcal{I}(\varphi))$ will be semantically equivalent to $\varphi$ even if they are not syntactically identical. Thus, we only need to check if $\varphi \equiv \varphi'$. For example, let $\varphi_0 = p_1 \wedge p_1$, $\psi_0 = \mathcal{I}(\varphi_1) = $ "*A conjunction of propositions $p_1$ and $p_1$ that can be simplified to $p_1$ using Idempotence.*", and $\varphi'_1 = \mathcal{A}(\psi_0) = p_1$ for a sequence $(\mathcal{A} \circ \mathcal{I})^1(\varphi_0)$. It is challenging to check if $\psi_0$ accurately represents $\varphi_0$, but it is easy to check if $\varphi_0 \equiv \varphi_1$ using a formal verifier.

Since ∀uto∃∨∧L uses formal syntax $\varphi$ as input and produces formal syntax $\varphi'$ as output, we can use formal verifiers to check whether $\varphi \equiv \varphi'$. As a result, ∀uto∃∨∧L avoids brittle syntactic equivalence checks and exhaustive tests of semantic equivalence that require evaluations of all possible truth valuations of formulas or executions of regexes.

We use the above insights to automatically assess LLM truth maintenance by using the same LLM $L$ to represent $\mathcal{I}$ and $\mathcal{A}$ respectively. Fig. 1 shows our overall assessment process. Briefly, we use a context-free grammar $\mathcal{G}$ to automatically generate a ground-truth *FL* expression $\varphi_0$. Next, we use a vocabulary generation process to generate a context for $\varphi_0$. This can either use abstract terms or use *NL* elements for more human-like scenarios (§D1). We then evaluate $(\mathcal{A} \circ \mathcal{I})^1(\varphi_0)$ by using $L$ to first generate $\psi_0 = \mathcal{I}(\varphi_0, \kappa)$ using context $\kappa$ designed for informalization. The context of $L$ is cleared (note that we only use the output of $\mathcal{I}(\varphi_0)$), and we use $L$ to generate $\varphi_1 = \mathcal{A}(\psi_0, \kappa')$ using context $\kappa'$ designed for autoformalization. We then use a verifier (e.g., Z3 (de Moura & Bjørner, 2008), Prover9 (McCune, 2010)) to assess if $\varphi_0 \equiv \varphi_1$ since both are elements of *FL*. If $\varphi_0 \equiv \varphi_1$ then we can repeat the process by evaluating $(\mathcal{A} \circ \mathcal{I})^1(\varphi_1)$ similarly.

*Example:* Consider case 2 in Fig. 1. ∀uto∃∨∧L uses the grammar in Fig. 2b to automatically generate a ground truth *FL* sentence as $\varphi_0 = p_1 \wedge p_2 \wedge p_1$. We can use any vocabulary to generate meaning for the propositions; $p_1$ : *It is raining today*, $p_2$ : *It was sunny yesterday*. Next, the LLM $L$ is prompted with Prompt 1 to perform informalization yielding *NL* $\psi_0 = \mathcal{A}(\varphi_0)$. $L$ can perform any simplification or other paraphrasing necessary. For example, $L$ could informalize $\varphi_0$ above to $\psi_0 = $"*The weather status was sunny yesterday whilst it is raining today.*" Notice that the LLM-generated *NL* statement automatically reflects a simplification using the Commutative $(a \wedge b \equiv b \wedge a)$ and Idempotent $(a \wedge a \equiv a)$ properties. Next, $L$ is asked to autoformalize $\psi_0$ without any context other than the vocabulary to use and a prompt for autoformalization (App. F). In this case, the LLM could return $\varphi_1 = \mathcal{A}(\psi_0) = p_1 \wedge p_2$. We use a theorem prover such as Prover9 (McCune, 2010) to show that $\varphi_0 \equiv \varphi_1$ and thus assess $L$'s truth maintenance capabilities w.r.t. $(\mathcal{A} \circ \mathcal{I})^1(\varphi_0)$.

## 3.2 ∀UTO∃∨∧L METRICS

**∀uto∃∨∧L score** When evaluating an LLM's truth-maintenance capabilities, it is crucial to consider the intended application, because performance on FL strings of similar complexity typically indicates how the model will fare in practice. As such, ∀uto∃∨∧L can be used in two distinct modes: parameterized and calibrated ∀uto∃∨∧L scores. The *parameterized ∀uto∃∨∧L score* computes performance with the descriptional complexity of FL strings as a parameter (e.g., the number of operators). The *calibrated ∀uto∃∨∧L score* $S_{cal}(D, d)$ is computed using all FL strings from dataset $D$ with complexity up to $d$, where there are equal number of examples for each complexity. In both modes, the score is computed as the fraction of FL strings in the corresponding dataset for which $(\mathcal{A} \circ \mathcal{I})^1(\varphi_1) \equiv \varphi_1$.

**Bounding false positives in computation of ∀uto∃∨∧L scores** A key advantage of the ∀uto∃∨∧L score is its robustness to different informalizations of the same *FL*. Thus, when ∀uto∃∨∧L outputs that an LLM maintains truth $(\mathcal{A} \circ \mathcal{I})^n(\varphi_0)$ on *FL* $\varphi_0$, the intermediate $NL = \mathcal{I}(\psi_0)$ is a semantically equivalent translation of $\varphi_0$. We now bound the probability of false positives, i.e., cases where the LLM fails both autoformalizing and informalizing but yields an FL string equivalent to the original.

Given an LLM $L$, let $\varphi_0 \xrightarrow{\mathcal{I}_L(\varphi_0)} \psi_0 \xrightarrow{\mathcal{A}_L(\psi_0)} \varphi_1$ be an execution of the ∀uto∃∨∧L process for $(\mathcal{A} \circ \mathcal{I})^1(\varphi_0)$ s.t. $\varphi_0 \equiv \varphi_1$ but $\psi_0$ is not an accurate representation of $\varphi_0$. We can derive the probability of ∀uto∃∨∧L providing such false positives. Let $p_{\mathcal{I}}$ be the probability with which $L$ informalizes an *FL* expression $\mathcal{I}(\varphi_0) = \psi_0$ s.t. $\psi_0$ is an accurate representation of $\varphi_0$. Similarly, let $p_{\mathcal{A}}$ be the probability of autoformalizing $\psi_0$, $\mathcal{A}(\psi_0) = \varphi_1$, s.t. $\varphi_1$ is semantically equivalent to $\psi_0$, i.e. $\varphi_0 \equiv \varphi_1$. Let $p_H$ be the probability that $L$ hallucinates *FL* $\varphi_1$ by autoformalizing $\psi_0$ s.t. $\varphi_1 \equiv \varphi_0$ given that $\psi_0$ is not an accurate representation of $\varphi_0$.

It can be seen that for a false positive to be outputted by ∀uto∃∨∧L, the sequence $\varphi_0 \rightarrow \psi_0$ produces an incorrect *NL* description and the sequence $\psi_0 \rightarrow \varphi_1$ autoformalizes incorrectly but hallucinates just right to yield $\varphi_1 \equiv \varphi_0$. The probability of such a sequence corresponds to $L$ making two mistakes, with the second mistake being such that it generated an expression equivalent to $\varphi_0$. This can be expressed as $(1 - p_{\mathcal{I}})(1 - p_{\mathcal{A}})p_H$. For $(\mathcal{A} \circ \mathcal{I})^n(\varphi_0)$, this probability is $(1 - p_{\mathcal{I}})^n(1 - p_{\mathcal{A}})^n p_H^n$ since ∀uto∃∨∧L computes $(\mathcal{A} \circ \mathcal{I})^1(\varphi_i)$ if $\varphi_{i-1} \equiv \varphi_i$ (Sec. 3). As LLM technology improves, we expect $p_{\mathcal{I}}, p_{\mathcal{A}} \rightarrow 1$ and $p_H \rightarrow 0$. As a result, the probability of false positives provided by ∀uto∃∨∧L decreases as $n$ increases. This low likelihood of false positives is further confirmed empirically by our analysis of correlation and predictive power w.r.t. other benchmarks (Sec. 4).

**LLMs as verifiers score** The llm-verifier score evaluates a given llm's ability to determine equivalence between FL strings. It is measured by using FL strings produced by a LLM from the ∀uto∃∨∧L process. For each dataset and descriptional complexity, we compute an $F_1$ score by comparing the evaluated LLM's equivalence predictions with the formal verifier's results. We use Chain-of-Thought (CoT) to allow LLMs to utilize their generated outputs to improve their reasoning (Wei et al., 2022).

**Predictive power** In addition to using calibrated and parameterized ∀uto∃∨∧L scores for assessing the ability for truth maintenance, we propose a new metric for evaluating the extent to which performance on a benchmark is indicative of performance on other benchmarks:

**Definition 3.1** (Predictive Power). Let $L_1$ and $L_2$ be language models evaluated on two benchmarks $A$ and $B$ with ranks $\geq_A$ and $\geq_B$. The *predictive power of A over B* is formally defined as $\mathcal{P}_{|A}(B) = Pr(L_1 \geq_B L_2 | L_1 \geq_A L_2)$.

In practice, we compute predictive power as a sampling-based maximum-likelihood estimate over multiple auto-generated datasets.

## 3.3 DYNAMIC DATASET GENERATION

We use context-free grammars (CFGs) (Hopcroft et al., 2001) – a set of *production rules* over *terminal* and *non-terminal* symbols – for dynamically generating datasets. An *infix parse tree* is obtained by repeatedly applying the rules, where the depth of this tree is often used to measure the *descriptional complexity* of a given string generated using the CFG (Csuhaj-Varjú & Kelemenová, 1993). CFGs also can be used to generate arbitrarily large amounts of data dynamically.

Another advantage is that CFGs can be customized with minimal human effort to generate diverse datasets whose ground-truth data possesses specific properties. For example, a dynamic dataset

$$S \to S \wedge S \qquad S \to (S \wedge S)|(S \vee S) \qquad S \to F|(\forall f.\, S)|(\exists f.\, S) \qquad S \to (S)K|\Sigma K$$
$$S \to (P \vee P \vee P) \qquad S \to (\neg S) \qquad F \to (F \wedge F)|(F \vee F) \qquad S \to S\Sigma K$$
$$P \to \neg v|v \qquad S \to \neg v|v \qquad F \to (\neg F)|\neg p|p \qquad K \to *|\varepsilon$$

| $(a)$ 3–CNF | $(b)$ Propositional Logic | $(c)$ First-order Logic | $(d)$ Regular Expression |
|---|---|---|---|

Figure 2: CFGs (described in Sec. 3.3.1) used for synthesizing the datasets in $\forall$uto$\exists\vee\wedge$L.

that only consists of $k$–CNF sentences – propositional logic in the Canonical Normal Form $(P_1^0 \vee \ldots \vee P_k^0) \wedge (P_1^1 \vee \ldots \vee P_k^1) \wedge \ldots$ where $P_i^j \in \{p_x, \neg p_x\}$ – can be easily generated. We enrich the generated sentence with context via a customizable *Vocabulary Generation* step, which automatically provides the necessary vocabulary for performing the task (e.g., providing English meanings to allow for human-like *NL*) by using terms from a vocabulary database or by using an LLM.

### 3.3.1 AUTO-GENERATED DATASETS

$\forall$uto$\exists\vee\wedge$L is open-source[1], is written in Python 3, includes several pre-computed datasets, and is easily customizable for adding new datasets, prompts, LLMs, etc. We now describe the datasets that any newly developed LLM can be evaluated on by using $\forall$uto$\exists\vee\wedge$L out-of-the-box.

Our dataset generator (described in App. B) takes a user-provided CFG and vocabulary to dynamically generate user-controlled, diverse datasets up to a user-specified metric such as the number of operators, parse tree depth, etc. It is guaranteed to generate any representable string using the CFG (App. B). As a result, users can easily generate out-of-distribution datasets in $\forall$uto$\exists\vee\wedge$L by simply providing CFGs and/or vocabularies.

The $\forall$uto$\exists\vee\wedge$L core benchmark uses four CFGs (Fig. 2) for producing five datasets compromising FL strings. The 3-CNF$(n)$ (Fig. 2a) and propositional logic PL$(n)$ (Fig. 2b) CFGs replace the terminal by randomly selecting from a list of $n$ propositions. First-order logic FOL$(n_p, n_o)$ (Fig. 2c) CFG replaces the terminal with predicates of the form $p(v_1, \ldots, v_n)$ where $p$ is a predicate name selected from a list of $n_p$ predicates, $v_i$ is either an object $o$ from a list of $n_o$ objects or is a free variable $f \in \{x_1, x_2, \ldots\}$ that is appropriately annotated within the scoping rules. Finally, the regular expression RE$(n)$ (Fig. 2d) CFG uses the vocabulary set $\Sigma = \{0, \ldots, n-1\}$.

We provide 5 datasets with 2 generated from the FOL CFG and 1 each for the rest. We sampled 500 strings for each complexity level. The 3-CNF(12) dataset contains examples with up to 59 operators, totaling $\sim 10k$ strings. PL(12) contains examples with up to 40 operators, for a total of $\sim 20k$ strings. The RE(2) dataset contains examples with tree depth up to 40, also totaling $\sim 20k$ strings.

The FOL datasets, FOL$(8, 12)$–S and FOL$(8, 12)$–E, contain examples with up to 37 operators, for a total of $\sim 19k$ strings each. FOL$(8, 12)$–S uses auto-generated synthetic object and predicate names. Conversely, FOL$(8, 12)$–E uses verbs from VerbNet (Schuler, 2005) for predicate names and names from Faker (Faraglia, 2024) for object names. Using more descriptive names allows for informalization to produce more *abstract* sentences that closely resemble the *NL* statements in SOTA autoformalization datasets. For example, a *FL* statement *Boom(Richard) $\wedge$ Exercise(Yolonda)* yields a more natural *NL* statement: *"Richard experiences a boom, and Yolonda engages in exercise"*.

While each dataset was generated in 10 separate pieces, each produced independently, our datasets contain $\sim 85k$ unique examples. We also provide zero-shot and 2-shot prompts for each dataset, for a total dataset size $\sim 170k$ for off-the-shelf evaluation and continual assessment of any new LLM. Of these examples, $\sim 85\%$ of them are composed of unique CFG parse trees (trees obtained by sampling the CFG but not injecting the vocabularies). Expressions with the same parse tree but different vocabularies (e.g., $p_1 \wedge p_2$ and $p_2 \wedge p_1$) account for $\sim 10\%$ of our dataset, providing a robust check against positional bias in the LLM. Additional information is presented in App. M.

We use open-source libraries to robustly parse the LLM-generated output. We use the Natural Language Toolkit (NLTK) library (Bird et al., 2009) for parsing logic and use Reg2Dfa (Reg, 2017) for regexes. LLM output that cannot be parsed is said to be *syntactically non-compliant*. Additionally,

---

[1]The code for this project is available at: `https://github.com/AAIR-lab/autoeval`.

---

Prompt 1: Informalization ($\mathcal{I}$) prompt for Fig. 1: Case 2 (other prompts available in App. F)

Your task is to convert a ⟨Propositional Logic, First-order Logic⟩ formula, appearing after [FORMULA], to a natural description that represents the formula. Only natural language terms are allowed to be used and do not copy the formula in your description. Your description should allow one to reconstruct the formula without having access to it, so make sure to use the correct names in your description. Explicitly describe the predicates. You may use terms verbatim as specified in the vocabulary below.

[VOCABULARY]
| | |
|---|---|
| Operators: | List of operators followed by their NL interpretations |
| Objects: | The objects in the universe (if any) |
| Propositions: | The propositions in the universe and their NL interpretations (if any) |
| Predicates: | The predicates in the universe and their NL interpretations (if any) |
| Examples: | Few-shot examples of the task (if any) |

- - - - - - - - - - - - - - - - - - - - - - - - - - - - - - - - - - - - - - - - - -

*Example Prompt*
Your task . . .
| | |
|---|---|
| Operators: | $\wedge$ represents conjunction, $\vee$ represents disjunction, . . . |
| Propositions: | $p_1$ : *It is raining*, $p_2$ : *It was sunny yesterday* |
| Formula: | $p_1 \wedge p_2 \wedge p_1$ |

- - - - - - - - - - - - - - - - - - - - - - - - - - - - - - - - - - - - - - - - - -

*Example Response:* `The sun was bright the day before whilst it is raining today.`

---

we also use scripts to ensure that the informalization step does not copy elements of *FL* into *NL* (e.g., complete or any parts of *FL*) that would otherwise make autoformalization trivial.

## 4 ASSESSMENT OF SOTA LLMS ON THE ∀UTO∃∨∧L BENCHMARK

In this section we present an evaluation of several SOTA LLMs using ∀uto∃∨∧L, as well as an evaluation of ∀uto∃∨∧L as a benchmark for evaluating LLMs' reasoning and translation ability using the predictive power score. In particular, we use the following assessment criteria for evaluating LLMs using ∀uto∃∨∧L: *(A1) Can LLMs produce FL translations that are syntactically compliant? (A2) Can LLMs maintain truth while translating FL? (A3) Can LLMs accurately verify whether two FL strings are logically equivalent?* In addition, we use the following criterion to assess ∀uto∃∨∧L itself: *(A4) Is the performance on ∀uto∃∨∧L indicative of performance on other benchmarks?*

### 4.1 EVALUATING LLMS USING ∀UTO∃∨∧L

We assessed §A1 - §A3 using 17 SOTA closed and open-source LLMs (Fig. 3). For clarity, we plot select models, grey out the data from the others, and refer the reader to App. N for a comprehensive overview. We evaluated §A1 and §A2 using the parameterized ∀uto∃∨∧L score on our generated datasets. For §A3, we calculated the LLMs as verifiers score for each descriptional complexity class by having each LLM verify the results produced by GPT-4o.

As stated in Sec. 2, prompts are crucial for LLM performance. To ensure our results reflect LLM capabilities rather than the effect of poorly designed prompts, we conducted extensive prompt engineering and ensured that at least one LLM could achieve a parameterized ∀uto∃∨∧L score $\geq 95\%$ on the 3-CNF(12) dataset, which has a constrained but representative grammar. Analysis on each LLM's performance on the 3-CNF(12) dataset is presented in App. C.

As shown in Fig. 3, SOTA LLMs are able to produce syntactically compliant formal syntax (§A1) for formal syntax with low descriptional complexity (e.g., few operators in logic). However, as the complexity increases, the ability of LLMs to autoformalize their own informalizations diminishes. One surprising result here is that GPT-4o is less syntactically compliant for regexes than Phi and LLama-3, which are much smaller models. This is due to GPT-4o often repeating a token sequence

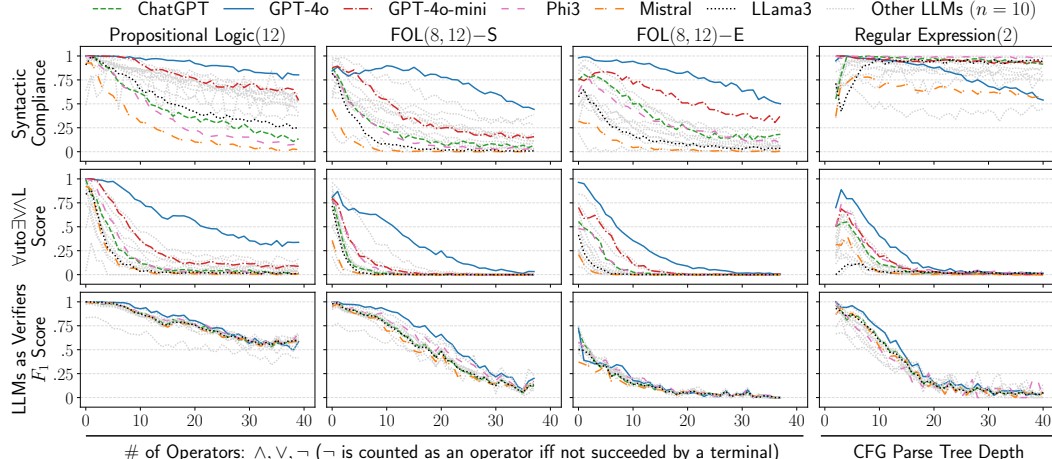

Figure 3: Zero-shot Pass@1 results from using ∀uto∃∨∧L to assess LLMs w.r.t. §A1, §A2, §A3 on the packaged datasets (Sec. 3.3.1). The x-axis represents an increasing descriptional complexity. The y-axis is each evaluated LLM's syntactic compliance rate (1st row), parameterized ∀uto∃∨∧L score (2nd row), and $F_1$ score as a verifier (3rd row). Additional results (prompt calibration, few-shot, etc.) are included in the Appendix.

when translating regex, resulting in hitting the token limit. For logic, we observed that LLMs often use the correct syntax but often misplace parentheses, creating malformed expressions.

Our analysis further shows, except on the 3-CNF(12) dataset used for prompt calibration, LLMs cannot maintain truth in FL translation (§A2) as the descriptional complexity increases. For translating logic expressions with more than 20 operators, none exceeded 50% accuracy in maintaining truth. This is concerning since formal specifications often have hundreds of operators. A common issue was misunderstanding the formal syntax's precedence and associativity rules. Misplaced operators led to quick verification failures. We provide an analysis of failing cases in App. G.

Moreover, even with CoT prompting, LLMs cannot serve as accurate verifiers of logical equivalence (§A3) for anything but toy expressions (low descriptional complexity), after which $F_1$ scores fall sharply. For small FL strings, we found that LLMs have difficulties with negations in logic. Due to space limitations, we present some examples and an analysis of the kinds of syntactic structures that LLMs fail to verify correctly in the Appendix (App. L, Fig. 21).

## 4.2 Evaluating ∀uto∃∨∧L as a Benchmark

For assessing §A4, we used the same 17 LLMs to evaluate the predictive power (Sec. 3.2) of ∀uto∃∨∧L w.r.t 5 popular benchmarks: (a) FOLIO(R;{NL, FOL}) (Han et al., 2024), a popular logical reasoning benchmark with ground truth in both *NL* and *FL*; (b) FOLIO({$\mathcal{A}/\mathcal{I}$}) evaluates if an LLM can (auto/in)formalize *NL* (*FL*) accurately; (c) LogiEval(R;{PL, FOL}) (Patel et al., 2024) a reasoning benchmark with ground truth in propositional and first-order logic; (d) HumanEval($\mathcal{A}$) (Chen et al., 2021), a code autoformalization benchmark; (e) Big Bench Hard (BBH) (Suzgun et al., 2023). These benchmarks are contrasted in Sec. 5, and example prompts of these benchmarks are included in App. K. We ran 5 runs on each benchmark except BBH. For BBH, we use the reported numbers in the literature as scores for the models (sources are included in App K). We measured the correlation between each benchmark's score and the calibrated ∀uto∃∨∧L score (Fig. 4), which was calibrated based on the descriptional complexity of the examples found in the benchmark. We also measured the calibrated ∀uto∃∨∧L score's predictive power w.r.t these benchmarks (Fig. 5).

As shown in Fig. 4, there is a moderate-to-strong positive correlation between LLM performance on ∀uto∃∨∧L and other logic-based benchmarks on a myriad of tasks such as autoformalization, logical reasoning, code generation, etc. The calibrated ∀uto∃∨∧L score exhibits a strong, positive correlation ($\rho \geq 0.7$) with other static benchmarks on *FL*-based tasks, as well as reasoning tasks such as FOLIO. Notably, calculating the parameterized ∀uto∃∨∧L score does not require hand-annotation unlike these benchmarks. Similar results appear in LogiEval for propositional logic, though the FOL version shows only a moderate correlation ($0.5 \leq \rho < 0.7$). We traced this reduction to dataset imbalance, where 80% of samples are from the positive class. Furthermore, the dataset is skewed

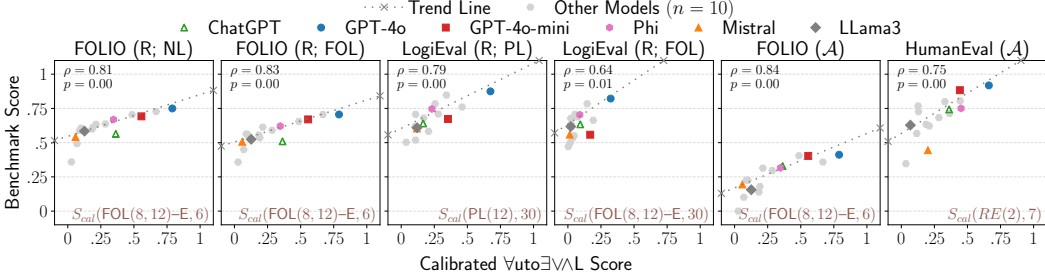

Figure 4: Correlation between scores on ∀uto∃∨∧L and static benchmarks from the literature. The Pearson correlation coefficient ($\rho$) and the $p$-value (values $\leq 0.05$ are statistically significant) are annotated in the top left. The calibrated ∀uto∃∨∧L score $S_{cal}(D, d)$ use all strings in dataset $D$ with descriptional complexity $d$ bounded above, as shown in the plots (App. K.4). Grey hexagons (⬡) represent data from 10 other models.

towards lower difficulty. This leads to lower overall performance (and consequently correlation) of models like GPT-4o-mini that actually try to reason and provide no answers compared to models like LLama-3.1-8b, which mostly answered yes.

Our results (Fig. 5) show that an LLM's calibrated ∀uto∃∨∧L score is a strong predictor of its performance on *FL*-based benchmarks. Our metric is also a more robust truth maintenance measure than length-dependent, *NL*-based metrics like BLEU scores (Papineni et al., 2002). For example, changing the generated *NL* $\psi =$ *"the weather status was sunny yesterday and is raining today"* to $\psi' =$ *"the weather status was sunny yesterday and is __not__ raining today"* still achieves a high BLEU($\psi', \psi$) score of 0.74 (BLEU($\psi, \psi$) = 1) but does not maintain truth. Even as a predictor for such metrics, ∀uto∃∨∧L notably surpasses random-chance accuracy.

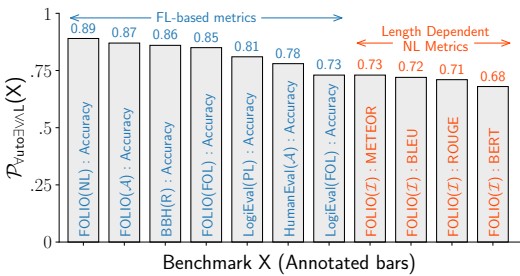

Figure 5: Predictive power of ∀uto∃∨∧L w.r.t other benchmarks. Benchmark metrics appear after the colon.

### 4.3 EVALUATING LARGE REASONING MODELS USING ∀UTO∃∨∧L

Large Reasoning Models (LRMs) are LLMs that also perform some reasoning steps (e.g., search) as a part of their generation process. We assessed two SOTA LRMs – OpenAI's o1 (OpenAI, 2024) and DeepSeek's R1 (DeepSeek, 2024) – on §A1 and §A2 using ∀uto∃∨∧L. Due to cost limitations, we regenerated a small dataset with 10 examples for each operator number for approximately 400 total examples. Our results (Fig. 6) show that even SOTA LRMs cannot maintain truth effectively in $(\mathcal{A} \circ \mathcal{I})^1(\varphi_0)$.

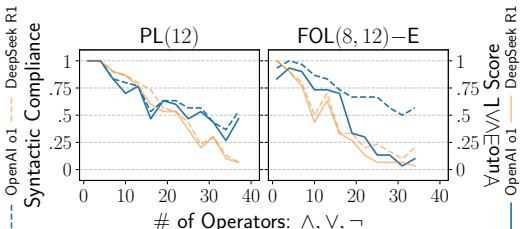

Figure 6: Applying ∀uto∃∨∧L with zero-shot prompts to LRMs on a small dataset of 400 strings.

## 5 RELATED WORK

**Logical Reasoning** RuleTaker (Clark et al., 2020) and ProntoQA (Saparov & He, 2023) generate datasets by using simple "if-then" and syllogisms rules to create reasoning questions. Similar grammars are used by LogicNLI (Tian et al., 2021) and CLUTRR (Sinha et al., 2019). LogiEval (Patel et al., 2024) uses fixed inference rules and LLMs to generate reasoning problems. Although these techniques are dynamic, they remain limited in generating interesting reasoning problems across different domains. In contrast, ∀uto∃∨∧L is multi-dimensional, offering five distinct datasets, multiple customization options, and the ability to produce an infinite number of unique syntax trees.

FOLIO (Han et al., 2024) utilizes human experts to generate a set of reasoning questions based on real-world text sources. They generate questions in both *NL* and *FL* for propositional and first-order logic that require 7 levels of reasoning. A similar approach is employed by ReClor (Yu et al., 2020) and (Srivastava et al., 2023). A key weakness of these approaches is their reliance on human experts.

**Autoformalization** HumanEval is a popular benchmark for evaluating LLM capabilities of autoformalizing source code. LLM autoformalizations are evaluated via hand-written test cases. It has been shown by Liu et al. (2023) through the HumanEval+ dataset that the test cases in HumanEval are incomplete and can provide misleading rankings. StructuredRegex (Ye et al., 2020) used crowdsourcing for generating regex datasets. In contrast, ∀uto∃∨∧L requires no human annotations and utilizes formal verifiers for checking the truth maintenance and thus does not share such drawbacks.

FOLIO($\{\mathcal{A}, \mathcal{I}\}$) (Han et al., 2024) tests the (auto/in)formalization abilities of LLMs by using hand-coded annotations of $\langle NL, FL \rangle$ pairs. However, as noted by the authors, they cannot check truth maintenance effectively and rely on an inference engine to compute truth values for each conclusion. ∀uto∃∨∧L uses theorem provers to check equivalence and thus is sound in its accuracy evaluation.

MALLS (Yang et al., 2024) is an autoformalization dataset for first-order logic that was generated using GPT-4. Their use of LLMs for generating the data limits the diversity of the dataset since and the authors suggest to only use this dataset for fine-tuning and not for evaluation. In contrast, ∀uto∃∨∧L generates correct *FL* and has a sound evaluation metric for truth maintenance.

Autoformalization approaches such LeanEuclid (Murphy et al., 2024), DTV (Zhou et al., 2024), LINC (Olausson et al., 2023), SatLM (Ye et al., 2020), Logic-LM (Pan et al., 2023) and others (Wu et al., 2022) utilize formal verifiers to provide sound evaluation metrics but utilize hand-coded datasets that limit their use in evaluating newer LLMs unlike ∀uto∃∨∧L.

**Informalization** Wu et al. (2022) and ProofNet (Azerbayev et al., 2023) use static datasets to evaluate LLM informalization capabilities. They use metrics such as BLEU scores that are known to not be indicative of accuracy for *FL*-based tasks (Ren et al., 2020). Jiang et al. (2023) develop MMA, a dataset of formal and informal pairs generated using GPT-4. They note that their dataset is an approximate measure due to using LLMs without manual validation. In contrast, ∀uto∃∨∧L is autonomous and provides sound measures of LLM capabilities w.r.t. truth maintenance.

## 6 CONCLUSION

We introduced ∀uto∃∨∧L, a new benchmark for autonomously assessing LLM truth maintenance in formal language translation. ∀uto∃∨∧L allows scalable data generation without human labeling and autonomously evaluates truth maintenance using formal verifiers to guarantee correctness. It is easily extensible and provides several prepackaged datasets and dataset generators to assess new LLMs quickly. Furthermore, our evaluation indicates that SOTA LLMs and LRMs are not performant in this task. Finally, we show that our metric is predictive of performance on other formal-language-based tasks and thus can be used as a surrogate benchmark for evaluating future LLMs.

**Broader Impact** ∀uto∃∨∧L provides a robust framework for evaluating the suitability and safety of LLMs in *FL*-based tasks such as autoformalization and code generation. It also serves as a surrogate for estimating performance as LLMs emerge. Our work lays the foundation for developing autonomous evaluation techniques for LLMs in more flexible syntaxes, such as conversational AI.

**Limitations and Future Work** A limitation of our work is the use of formal verifiers: the equivalence problem for first-order logic is well known to be undecidable. We mitigate this by using an appropriate timeout and logging (only $0.66\%$ of our results experienced a timeout). This issue can be removed by using CFGs that generate decidable strings. An interesting application of ∀uto∃∨∧L is using generated evaluations as datasets for back-translation, thereby improving the autoformalization capabilities of models (Jiang et al., 2023). One interesting extension of our would be assessing truth maintenance in other applications (e.g., image generation). Finally, using formal verifiers as callable tools for an LLM is an intriguing extension of our benchmark, enhancing the assessment of §A3.

**Threats to Validity** Our results for paid APIs rely on the model checkpoints to be available. Due to cost limitations, our results report pass@1 scores. We do report the std. deviation across 10 runs on 10% of each dataset in App. H with similar conclusions. We assume that the verifier and parsing libraries are sound. We mitigate this risk by using popular open-source tools like Prover9 and NLTK.

## ACKNOWLEDGEMENTS

This work was supported in part by the ONR grant N00014-23-1-2416, NSF grant IIS 1942856, the Open AI Researcher Access Grant, and Arizona State University's GPSA Jumpstart Research Grant. We acknowledge Research Computing at Arizona State University for providing computation resources that contributed to this paper's results.

## ETHICS STATEMENT

Our work involves using LLMs for generating text. Naturally, it is imperative to ensure that appropriate guardrails are in place to prevent offensive content from being generated and/or displayed. We do not use any personally identifiable information in $\forall$uto$\exists\lor\land$L.

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

## A    APPENDIX ORGANIZATION

The code used for this project can be found at `https://github.com/AAIR-lab/autoeval`.

The Appendix is organized as follows. Appendix Appendix B provides the algorithm used for dataset generation. Appendix C discusses prompt tuning and validating our prompts on 3–CNF. Appendix D provides the parameters we used when generating the five datasets discussed in the paper. Appendix E provides additional information on our experimental setup, including the computational resources used. Appendix F discusses the prompts and provides examples. Appendix G is our detailed analysis of the empirical results from the main paper. Appendix H discusses an experiment we ran to evaluate the standard deviation error. Appendix I includes additional results from our zero-shot prompting experiments using other metrics for categorization. Appendix J evaluates an experiment we performed comparing few-shot prompting compared to zero-shot. Finally, Appendix K provides the experimental setup of the benchmarks we evaluated, data values and sources of scores collected, the calibrated ∀uto∃∨∧L scores used for comparison, and additional correlation results.

## B    DATASET GENERATION

In this section, we provide the algorithm for generating formal syntax (FS) expressions and show that it can generate all possible expressions from the grammar and vocabulary.

Our approach, ∀uto∃∨∧L, generates datasets by constructing a context-free grammar (CFG) tree using the grammars discussed in Section 3.3.1. Since it is intractable to generate the full tree, we control the branching factor and randomly expand the branches of this tree to generate formulae.

---

**Algorithm 1** Dataset Generation

1: **Inputs:** CFG $\mathcal{G}$, vocabulary $\mathcal{V}$, branching factor $n$, tree depth *depth*, sample count *sample_count*, and categorization metric $m$.
2: **Outputs:** set of FS expressions $\overline{\varphi}$
3: $\mathcal{N} \leftarrow \{0 : [None]\}, \mathcal{N}_t \leftarrow \langle\rangle$
4: **for** $d = 1, 2, \ldots, depth$ **do**
5:     $\mathcal{N}' \leftarrow sampleN(\mathcal{N}[d-1], n)$
6:     **for** $\nu \in \mathcal{N}'$ **do**
7:         $\mathcal{N}_\nu, \mathcal{T}_\nu \leftarrow generateNChildren(\nu, \mathcal{G}, n)$
8:         $\mathcal{N}[d] \mathrel{+}= \mathcal{N}_\nu$
9:         $\mathcal{N}_t \leftarrow \mathcal{N}_t \cup \mathcal{T}_\nu$
10:     **end for**
11: **end for**
12: $M \leftarrow categorizeExpressionsIntoDict(\mathcal{N}_t, m)$
13: $\overline{\varphi} \leftarrow \langle\rangle$
14: **for** $k \in keys(M)$ **do**
15:     $M_k \leftarrow sampleCFGExpressions(M[k], sample\_count)$
16:     $\overline{\varphi}_k \leftarrow buildFSExpressions(M_k, \mathcal{V})$
17:     $\overline{\varphi} \leftarrow \overline{\varphi} \cup \overline{\varphi}_k$
18: **end for**
19: **Return:** $\overline{\varphi}$

---

The dataset generation algorithm is shown in Algorithm 1. This algorithm constructs a CFG tree by maintaining non-terminal nodes at each tree level ($\mathcal{N}$) and all the leaf nodes ($\mathcal{N}_t$), where each terminal node represents a completed CFG expression (line 3). For generating nodes at a certain level in the tree, $n$ nodes from the previous level are sampled (line 5). Each node is branched $n$ times using the CFG to produce nodes at the current tree level, and all the leaf nodes are collected (lines 7 through 9). As a result, by iteratively performing this process for each tree level, we obtain a set of leaf nodes (CFG expressions).

The leaf nodes are then categorized based on the specified metric (e.g., tree depth, number of operators, etc.) (line 12). For each metric value, a fixed number of CFG expressions corresponding to that value are sampled (line 15). Using the vocabulary, an FS expression is constructed from each CFG expression (line 16). Consequently, the final dataset of FS expressions contains an equal number for

each metric value (line 17). This set of FS expressions is the final result produced by the algorithm (line 19).

The vocabulary is fixed in length, with a hyperparameter controlling the number of unique propositions for propositional logic. Similarly, for first-order logic, the number of unique variables, constants, and predicates are also hyperparameters. Also, regular expressions have a hyperparameter controlling the alphabet size. When these expression components are needed for building the FS expression, the exact one is selected using uniform random selection. In the special case of first-order logic predicates, the grounded predicate is generated by randomly selecting a predicate and then selecting constants depending on the predicate's arity. In the case of the arbitrary vocabulary, the arity for a predicate is randomly assigned. To add variables, each constant has a certain probability of being replaced by a variable.

**Guaranteed Expression Coverage** The dataset generator (Algorithm 1) is guaranteed to generate all possible formal syntax expressions that can be produced for a grammar and vocabulary. Let $\varphi$ be an FS expression that can be constructed using the rules from CFG $\mathcal{G}$ and the vocabulary $\mathcal{V}$. Note that $\varphi$ corresponds to a CFG expression $\varphi_{CFG}$, derived by substituting the vocabulary with the CFG symbols. Due to uniform selection, the probability of $\varphi$ being generated from $\varphi_{CFG}$ is greater than zero. Furthermore, the CFG expression represents a leaf node in the CFG tree that can be reached by applying the CFG rules in a specific sequence. Due to the random sampling of rules at each node, there is a non-zero probability of generating this particular path in the tree. Thus, $\varphi$ can be generated using the dataset generator algorithm.

## C  3-CNF PROMPT CALIBRATION

In this section, we discuss the K-CNF results used to calibration the prompts.

We tested several prompts for 3-CNF to verify that our prompts are sufficient to prompt the LLM to correctly perform informalization and autoformalization. Additionally, we verified that the equivalence verification prompt prompted the LLMs to give an accurate yes-or-no answer. The performance of all 14 LLMs on 3-CNF for §A1, §A2, and §A3 are shown in Figure 7.

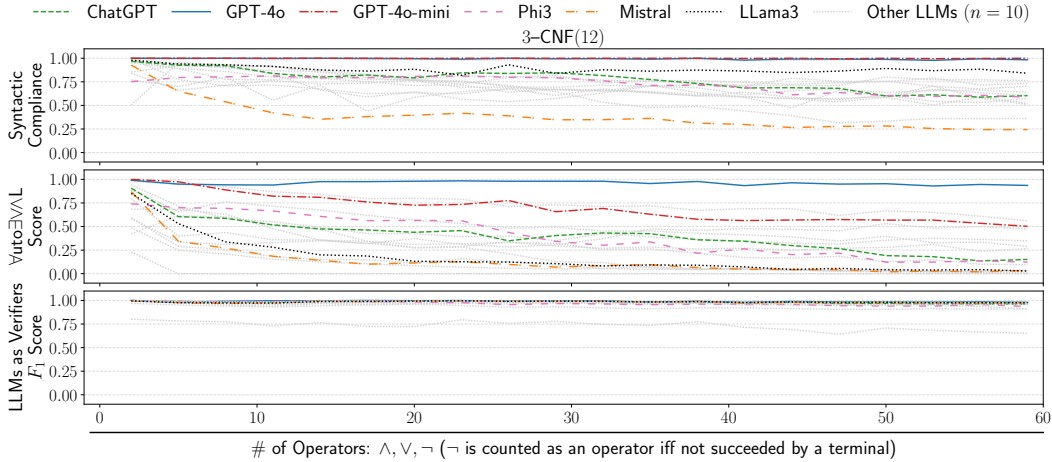

Figure 7: Zero-shot Pass@1 results (avg. over 10 batches, higher values better) for 3-CNF from using ∀uto∃∨∧L to assess LLMs w.r.t. §A1, §A2, §A3 (Sec. 3.3.1) on the packaged datasets. The x-axis is the # of operators.

The best-performing models we tested (GPT-4o and GPT-4o-mini) achieved nearly perfect syntactic compliance, accuracy, and equivalence verification even as the number of operators increased. This proves that the prompts we used in our experiments are sufficient for prompting the model for performing the tasks for §A1, §A2, and §A3.

For the other LLMs tested, syntactic compliance and accuracy diminished as the number of operators increased. However, when evaluating the equivalence of GPT-4o results, all LLMs achieved near-perfect accuracy regardless of operator number. Due to most of GPT-4o results being positive cases, the results support that LLMs can verify two equivalent 3-CNF formulae as equivalent.

## D  DATASET GENERATION HYPERPARAMETERS

In Table 1, we provide the hyperparameters used to generate the five datasets.

Table 1: Hyperparameters used for producing the five datasets.

| Parameter Type | Hyperparameter | Value | Description |
|---|---|---|---|
| **General** | `depth` | 40 | Maximum depth of the CFG tree. |
| | `n` | 200 | Branching factor of produced CFG tree. |
| | `sample_count` | 50 | Number of CFGS for each metric value to select. |
| **First-Order Logic** | `free_variable_prob` | 0.25 | Probability of a constant being replaced by a variable. |
| | `max_free_variables` | $\infty$ | Maximum number of unique variables. |
| | `max_predicate_arity` | 2 | Maximum predicate arity. |
| | `min_predicate_arit` | 1 | Minimum predicate arity. |
| | `num_objects` | 12 | Number of unique constants. |
| | `num_predicates` | 8 | Number of unique predicates. |
| **Propositional Logic** | `num_propositions` | 12 | Number of unique propositions. |
| **Regular Expression** | `alphabet_size` | 2 | Alphabet size. |

## E  EXPERIMENTAL SETUP

In this section, we will provide the details of our experimental setup for generating the datasets and running ∀uto∃∨∧L for evaluating each LLM's performance.

We ran our experiments using Python 3.10.13 with package versions shown in Table 2. We also repackaged Prover9 (McCune, 2010) to improve performance where this repackaged version can be found in our code base.

**Dataset Generation:** We generated five datasets using the dataset generation algorithm with the hyperparameters shown in Table 1 using the number of operators as the categorization metric for all but regular expression, where we used CFG tree depth. We generated 10 batches for each dataset, resulting in approximately 20k samples for each dataset with an equal distribution for each operator number.

**Evaluating and Verification:** The closed-source models (GPT-3.5-turbo, GPT-4o, and GPT-4o-mini) were accessed using their API using a temperature of 0.1. The open-source models LLama-3-8B-Instruct and Mistral-v0.2-7B-Instruct were locally hosted on a server with a 13th Gen Intel(R) Core(TM) i9-13900K and Nvidia RTX 4090 GPU using the model's default parameters with a temperature of 1. Similarly, Phi-3-medium-4k-instruct was locally hosted on a server using a Nvidia A100-XM4-80GB GPU.

Table 2: Python package versions used for empirical evaluation.

| Python Package | Version |
|---|---|
| `openai` | 1.45.0 |
| `nltk` | 3.8.1 |
| `tqdm` | 4.66.4 |
| `anthropic` | 0.26.1 |
| `backoff` | 2.2.1 |
| `tiktoken` | 0.6.0 |
| `transformers` | 4.41.1 |
| `Faker` | 25.2.0 |
| `networkx` | 3.3 |

Verification was performed on an AMD EPYC machine with 128 cores. The larger open-source models, Yi-1.5-34B-Instruct and Llama-3-70B-Instruct, were run on Arizona State University's Sol supercomputer (Jennewein et al., 2023).

# F  PROMPTING

In this section, we provide the zero-shot and few-shot used in the main paper experiments.

The prompt for each dataset type provides the LLM with information on the problem type and the vocabulary. For informalization, we prompt the model to produce just a natural language description. We also provide the list of objects, predicates, propositions, and free variables in the formal syntax expression. For autoformalization, the LLM is prompted to provide just the formal syntax expression using the natural language description. Additionally, for first-order logic with a non-synthetic grammar, we provide the predicate names and arity in the autoformalization prompt. Example informalization and autoformalization few-shot prompts for the first-order logic dataset are shown in Prompt 2 and Prompt 3. Example informalization and autoformalization few-shot prompts for the regular expression dataset are shown in Prompt 4 and Prompt 5. Example informalization and autoformalization zero-shot prompts for the propositional logic dataset are show in Prompt 6 and Prompt 7.

For §A4, the prompt used for using an LLM to verify the equivalence of two formulae tells the LLM about the type of datasets (e.g., propositional logic, first-order logic, and regular expression). Using Chain-of-Thought prompting, the model is prompted to provide an explanation before giving a yes-or-no answer in a parsable format. Prompt 8 gives an example of the prompt we used for verifying two first-order logic formulae.

# G  ANALYSIS OF MAIN PAPER RESULTS

In this section, we analyze the main empirical results of the paper. Our results clearly show that current SOTA LLMs are not performant in the truth maintenance task, which is why $\forall$uto$\exists\lor\land$L is needed. As the expression complexity increases, the syntactic compliance, accuracy, and ability to verify equivalence diminishes. We describe some of the errors that cause the low accuracy for propositional logic, first-order logic, and regular expressions.

## G.1  PROPOSITIONAL LOGIC RESULTS

Table 3: Examples of errors the evaluated LLMs made while evaluating $\forall$uto$\exists\lor\land$L for the propositional logic dataset.

| $\varphi$ | $\mathcal{I}(\varphi)$ | $(\mathcal{A} \circ \mathcal{I})(\varphi)$ |
|---|---|---|
| $(\neg p11 \land \neg p8)$ | The natural description of the formula is: "It is not the case that both proposition p11 and proposition p8 are true." | $(\neg(p11 \land p8))$ |
| $(\neg p10 \land (\neg p9 \lor p7))$ | The natural description of the formula is: "It is not the case that proposition p10 is true and either proposition p9 is false or proposition p7 is true." | $\neg(p10 \land (\neg p9 \lor p7))$ |
| $(\neg p3 \land \neg p7)$ | Here is the natural description of the given formula: "The statement says that p3 is not true and p7 is also not true." | $(\neg p3 \lor \neg p7)$ |
| $(\neg\neg p2 \lor p3)$ | The formula states that either the proposition p2 is true or the proposition p3 is true. Additionally, it specifies that the proposition p2 is not false. | $(p2 \lor p3) \land \neg\neg p2$ |
| $(\neg p2 \land p5 \land \neg p6)$ | The formula describes a situation where the first proposition is false, the second proposition is true, and the third proposition is false. | $(\neg p \land q \land \neg r)$ |

**Informalization Errors:** A common error was the LLM failed to describe the proposition names. Another was the LLM failing to provide a complete description of the formula. For example, GPT-3.5-turbo often described portions of the expression based on what propositions and operators it

---

**Prompt 2: Few-Shot First-Order Logic Informalization Prompt**

[TASK]
Your task is to convert a first-order logic formula, appearing after [FORMULA], to a natural description that represents the formula. Only natural language terms are allowed to be used and do not copy the formula in your description. Your description should allow one to reconstruct the formula without having access to it, so make sure to use the correct names in your description. Explicitly describe the predicates. You may use terms verbatim as specified in the vocabulary below.

[EXAMPLE 1]
$(\neg p2 \vee p1 \vee \neg p2)$
Disjunctive predicate logic expression consisting of three components: the negation of a proposition labeled p2, the proposition p1, and again the negation of p2.

[EXAMPLE 2]
$(\neg\neg p2 \wedge \neg(p3 \vee p1))$
The expression asserts that p2 is not false while both p3 and p1 are not true.

[VOCABULARY]
$\vee$ represents disjunction
$\wedge$ represents conjunction
$\neg$ represents negation
( and ) represent parentheses
propositions can be used verbatim
predicates can be used verbatim
$\forall < x1 >< x2 > ... < xn >$ . represents universal quantification with x1... representing free variables
$\exists < x1 >< x2 > ... < xn >$ . represents existential quantification with x1... representing free variables
The objects are: $p5, x1$
The parameterized predicates are: $pred3(?p0, ?p1)$
The free variables are: $x1$

[FORMULA]
$\forall x1 \; pred3(p5, x1)$

---

contained. A common issue with GPT-4o, one of the best models, is that it often uses different propositional symbols (see example 5 in Table 3). Finally, we also observed hallucinations were the LLM attempted and failed to simplify the original formula (see example 4 in Table 3). These interpretation errors resulted in the original meaning of the expression being lost.

**Autoformalization Errors:** We observed there were often syntactic issues where the description was not fully translated into a formula or the parentheses did not match. An interesting result is that the LLMs struggled to place the negation operator in the correct location. For example, GPT-4o often describes $\neg p \wedge \neg p$ as predicate $p$ "negated twice and combined" but failed to regenerate the original formula properly with this description.

### G.2 First-Order Logic Results

**Informalization Errors:** Similar to propositional logic, we observed the LLM often failed providing enough details resulting in incorrect formulas being generated. A significant source of errors we observed when not providing the predicate names and arity was the LLM rephrasing its explanation causing confusion when regenerating.

**Autoformalization Errors:** Beyond the errors observed in propositional logic, the most common mistake made during autoformalization was the LLM confusing constants with variables (see example 2 in Table 4). Additionally, the LLMs often messed up the predicate arity. Mistral often used = and

---

**Prompt 3: Few-Shot First-Order Logic Autoformalization Prompt**

[VOCABULARY]
Use $\vee$ to represent disjunction
Use $\wedge$ to represent conjunction
Use $\neg$ to represent negation
Use ( and ) to represent parentheses
Use $\forall$ <free_variable_list> to represent universal quantification
Use $\exists$ <free_variable_list> to represent existential quantification
The <free_variable_list> consists of a sequence of space separate free variables with the last variable immediately followed by a period. Examples: (1) all x1 x2. (2) exists x4.
Use <predicate>(<parameter_list>) to represent predicates (Names and parameters are provided in the description)

[TASK]
Your task is to interpret the natural language (NL) description of a first-order logic formula and represent it as formal syntax using the vocabulary specified in the [VOCABULARY] block above. Only output the formula and no other text. The NL description appears immediately following the [NL DESCRIPTION] tag.

[EXAMPLE 1]
Disjunctive predicate logic expression consisting of three components: the negation of a proposition labeled p2, the proposition p1, and again the negation of p2.
$(\neg p2 \vee p1 \vee \neg p2)$

[EXAMPLE 2]
The expression asserts that p2 is not false while both p3 and p1 are not true.
$(\neg\neg p2 \wedge \neg(p3 \vee p1))$

[NL DESCRIPTION]
For all objects labeled x1, the predicate pred3 holds true with parameters p5 and x1.

---

Table 4: Examples of errors the evaluated LLMs made while evaluating $\forall$uto$\exists\vee\wedge$L for the first-order logic dataset.

| $\varphi$ | $\mathcal{I}(\varphi)$ | $(\mathcal{A} \circ \mathcal{I})(\varphi)$ |
|---|---|---|
| $(\neg pred8(p10) \wedge pred8(p5) \\ \wedge pred6(p8))$ | The formula states that it is not the case that the predicate pred8 holds for the object p10, and the predicate pred8 holds for the object p5, and the predicate pred6 holds for the object p8, all conjoined together. | $\neg(pred8(p10) \\ \wedge pred8(p5) \\ \wedge pred6(p8))$ |
| $\exists x1.\neg pred2(p4)$ | Here is the natural description of the given formula: "There exists an x1 such that p4 is not an instance of pred2. | $\exists x1.\neg pred2(x1)$ |
| $\forall x1.\neg\neg pred3(p5)$ | For all x1, it is not the case that pred3(p5) is false. | $\forall x1.\neg(pred3(p5) \\ \vee \neg pred3(p5))$ |
| $pred2(p3, p5)$ | There exists a relationship between the objects p3 and p5 such that the predicate pred2 holds true for these objects. | $\exists p3\ p5.pred2(p3, p5)$ |

$\neq$ operators with the variables, which was not needed for any formluae in $\forall$uto$\exists\vee\wedge$L. Similarly, the LLMs would often use their own grammar instead of the one provided in the prompt.

---

**Prompt 4: Few-Shot Regex Informalization Prompt**

[TASK]
Your task is to convert the regular expression appear after [REGEX], to a natural description that represents the regular expression. Only natural language terms are allowed to be used and do not copy the regular expression in your description. Your description should allow one to reconstruct the regular expression without having access to it, so make sure to use the correctly account for scoping. You may use terms verbatim as specified in the vocabulary below.

[VOCABULARY]
you may use symbols from the vocabulary
you can use *

[EXAMPLE 1]
(1*)0*
The regex matches strings that starts with any number (including none) of the digit '1', followed by any number (including none) of the digit '0'.

[EXAMPLE 2]
(01*)
The regex matches strings that begin with a '0' followed directly by any number (including none) of '1's.

[FORMULA]
0

---

**Prompt 5: Few-Shot Regex Autoformalization Formal**

[VOCABULARY]
Use * to represent zero or more duplications of the same expression
Use ( and ) to represent parentheses

[TASK]
Your task is to interpret the natural language (NL) description of a regular expression and represent it as formal syntax using the vocabulary specified in the [VOCABULARY] block above. Only output the regular expression and no other text. The NL description appears immediately following the [NL DESCRIPTION] tag.

[EXAMPLE 1]
The regex matches strings that starts with any number (including none) of the digit '1', followed by any number (including none) of the digit '0'.
(1*)0*

[EXAMPLE 2]
The regex matches strings that begin with a '0' followed directly by any number (including none) of '1's.
(01*)

[NL DESCRIPTION]
The regex matches strings that start with the digit '0'.

---

## G.3 REGULAR EXPRESSION RESULTS

**Informalization Errors:** Most of the errors observed were the LLMs giving the wrong explanation, even for simple regular expressions. For example, GPT-4o often described $c^*$ as "one or more

---

**Prompt 6: Zero-Shot Propositional Logic Informalization Prompt**

[TASK]
Your task is to convert a propositional logic formula, appearing after [FORMULA], to a natural description that represents the formula. Only natural language terms are allowed to be used and do not copy the formula in your description. Your description should allow one to reconstruct the formula without having access to it, so make sure to use the correct names in your description. Explicitly describe the predicates. You may use terms verbatim as specified in the vocabulary below.

[VOCABULARY]
$\vee$ represents disjunction
$\wedge$ represents conjunction
$\neg$ represents negation
( and ) represent parentheses
propositions can be used verbatim
The propositions are: p5, p12, p4

[FORMULA]
$(p5 \vee \neg p12 \vee \neg p4)$

---

**Prompt 7: Zero-Shot Propositional Logic Autoformalization Prompt**

[TASK]
Your task is to interpret the natural language (NL) description of a propositional logic formula and represent it as formal syntax using the vocabulary specified in the [VOCABULARY] block above. Only output the formula and no other text. The NL description appears immediately following the [NL DESCRIPTION] tag.

[VOCABULARY]
Use $\vee$ to represent disjunction
Use $\wedge$ to represent conjunction
Use $\neg$ to represent negation
Use ( and ) to represent parentheses

[NL DESCRIPTION]
A disjunctive statement involving three propositions: p5, the negation of p12, and the negation of p4.

---

occurrences of 'c'", where $c$ is a character from the alphabet (see example 1 in Table 5). For the other LLMs, it was quite common for the explanation to not give the actual character (see example 4 in Table 5). Overall, we observed a higher likelihood of SOTA LLMs hallucinating on regular expressions compared to the other datasets.

**Autoformalization Errors:** The most common mistake when constructing a regular expression from natural language was misplacing $*$ or adding it when it was not needed (see example 3 in Table 5). Finally, even though we explicitly prompted the LLMs to use just $*$, sometimes the LLM would use $+$.

## H   STANDARD DEVIATION EVALUATION

In this section, we perform an empirical analysis of the standard deviation of the syntactic compliance and accuracy of the $\forall$uto$\exists\vee\wedge$L results. Due to the 10 batches having different data, the standard deviation cannot be computed reliably based on the performance of the individual batches. We evaluated the standard deviation by running $\forall$uto$\exists\vee\wedge$L 10 times on the first batch of each dataset

> **Prompt 8: First-Order Logic Verification Prompt**
>
> Your task is to say whether two First-Order Logic formulae are equivalent. The first formula will appear right after [FORMULA 1] and the second after [FORMULA 2].
> Give an explanation followed by a yes or no answer. The answer must show up at the end with the format "[Answer]" followed by either a yes or no.
>
> [Formula 1]
> $\exists x1.\neg pred5(p7)$
>
> [Formula 2]
> $\exists p7.\neg pred5(p7)$

Table 5: Examples of errors the evaluated LLMs made while evaluating with $\forall$uto$\exists\lor\land$L for the regular expression dataset.

| $\varphi$ | $\mathcal{I}(\varphi)$ | $(\mathcal{A} \circ \mathcal{I})(\varphi)$ |
|---|---|---|
| $1^*0$ | The pattern consists of one or more occurrences of the digit "1" followed by exactly one occurrence of the digit "0". | $(1^*)10$ |
| $(1^*)^*0$ | The regular expression describes a sequence that consists of zero or more repetitions of the digit '1', followed by exactly one '0'. This entire pattern can repeat any number of times, including zero times. | $((1^*)0)^*$ |
| $1^*11^*$ | The pattern consists of zero or more occurrences of the digit '1', followed by exactly one occurrence of the digit '1', followed by zero or more occurrences of the digit '1'. | $1^*1^*1^*$ |
| $(1)^*0$ | Zero or more occurrences of the character or group of characters before the asterisk. | $(.^*)$ |

composed of 1974 propositional logic, 1900 first-order logic, and 1842 regular expressions examples. Additionally, we evaluated GPT-3.5-turbo (ChatGPT) with a temperature of 1, LLama-3-8B-Instruct, Mistral-v0.2-7B-Instruct, and Phi-3-medium-4k-instruct. We calculated the mean and standard deviation of each independent run of $\forall$uto$\exists\lor\land$L and plotted the results in Figure 8.

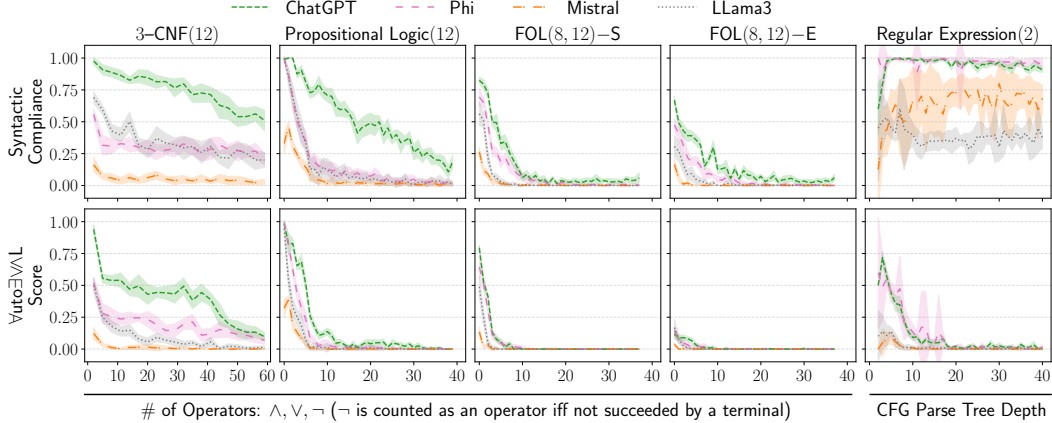

Figure 8: Average and standard deviation error of Zero-shot Pass@1 results from using $\forall$uto$\exists\lor\land$L to assess LLMs w.r.t. §A1 and §A2 (Sec. 3.3.1) on the first batch of the packaged datasets. The x-axis represents an increasing order of descriptional complexity.

For propositional and first-order logic, the standard deviation of the evaluated LLMs is low. While noisier, the standard deviation of the regular expression results were still less than 20% with the

better performing models having a lower standard deviation. Overall, this experiment shows that the noise of non-deterministic text generation does not significantly impact ∀uto∃∨∧L or our results and evaluations.

## I  ADDITIONAL ZERO-SHOT PROMPTING RESULTS

In this section, we evaluate other categorization metrics from the zero-shot prompting experiments from the main paper. For the propositional and first-order logic datasets, the other categorization metrics are the CFG parse tree depth needed to produce each FS expression and the individual number of each operator $(\wedge, \vee, \neg)$. For regular expressions, we have discussed in the main paper that each regular expression represents a minimal deterministic finite automata (DFA) that is unique up to isomorphism. Therefore, the other categorization metrics for regular expressions are the number of nodes $V$, the number of edges $E$, and the density of this minimal DFA. The density is calculated using Equation 1 where we discretize the value by rounding to every tenth.

$$Density = \frac{|E|}{|V|(|V| - 1)} \tag{1}$$

**Imbalanced Dataset Labels** Due to the datasets being created by sampling an equal number of expressions for each number of operators, taking this dataset and evaluating it in terms of the other metrics results in an imbalanced dataset. To examine this effect, we have created Figures 9 and 10 to perform an analysis of dataset imbalance on these other metrics.

For propositional and first-order logic, the dataset is actually quite balanced due to CFG tree depth and the number of each individual operator having a high correlation to the total number of operators. As such, other than metric values close to the extrema, the noise from the imbalanced data will be marginal.

The regular expression dataset is less balanced due to a weaker correlation with the CFG tree depth. The middle of the density graphs will be the most noisy since there is significantly less data for densities of 0.1 and 0.2. The number of examples drops as the number of edges and nodes increases with less than 10% of the data having more than 7 edges and/or nodes.

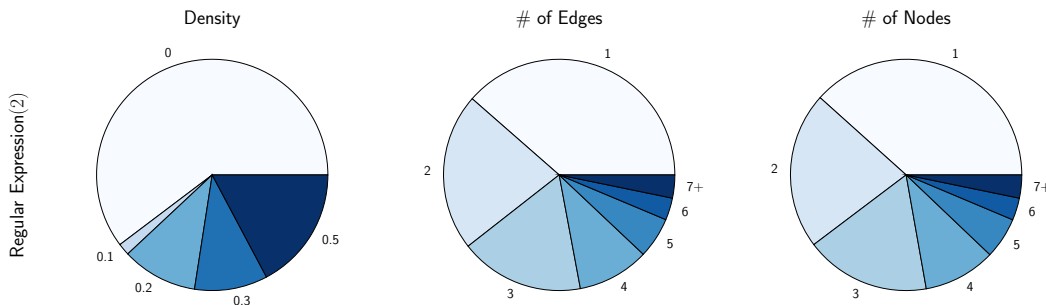

Figure 9: Count of the number of examples for each metric value for the regular expression datasets. The pie charts increase in values counter-clockwise while going from lighter to darker.

**Categorization Metrics Performance** In Figures 11, 12,13, 14, and 15 the performance of each LLM over these other categorization metrics are shown. Across the board, we observe a diminishing performance regardless of the source of increasing complexity. Ignoring the noise from the low number of examples closer to the extrema, the depth of the tree showed a similar behavior as the operator number. Propositional logic performance was concave w.r.t the number of $\wedge$ and $\vee$ operators since it becomes easier to describe expressions composed of exclusively $\wedge$ and $\vee$ operators. A similar, but weaker pattern is observed in the first-order logic results for the same reason. The negation operator was not concave, showing how LLMs struggle to handle multiple negation operators.

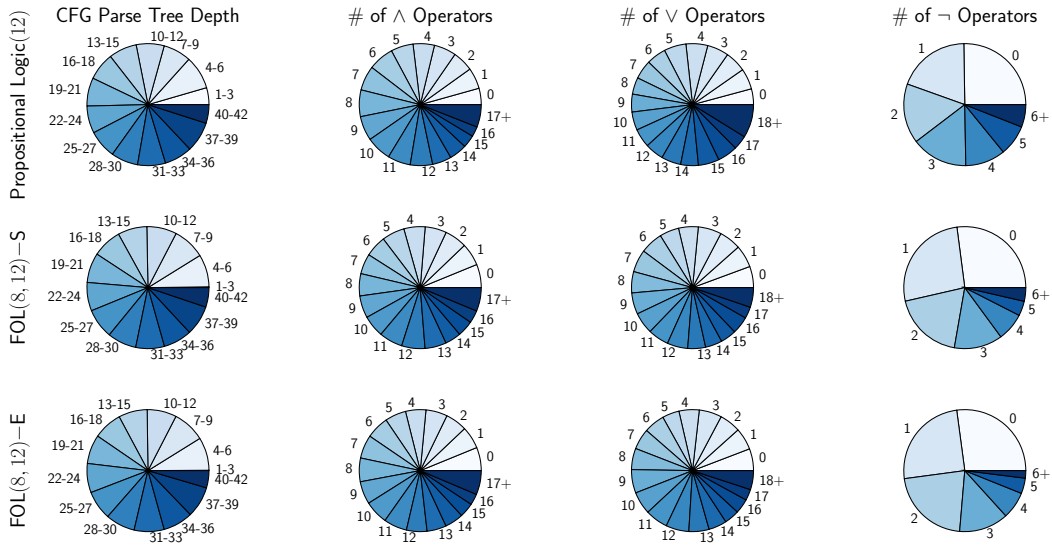

Figure 10: Count of the number of examples for each metric value for each of the datasets. Each row is a dataset and each column is a different metric that can be used to categorize the dataset. The pie charts increase in value counter-clockwise while going from lighter to darker.

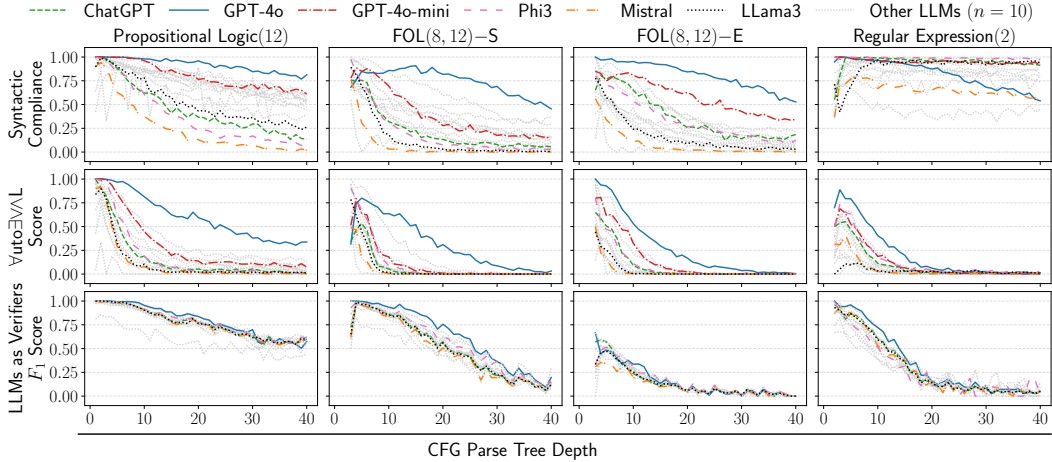

Figure 11: Zero-shot Pass@1 results (avg. over 10 batches, higher values better) from using ∀uto∃∨∧L to assess LLMs w.r.t. §A1, §A2, §A3 (Sec. 3.3.1) on the packaged datasets. The x-axis is the depth of the CFG tree to produce the formula.

For regular expressions, increasing the number of nodes and edges reduces accuracy and the ability to evaluate equality. Density does not seem to be a factor, as the dip at 0.1 can be associated with noise due to the lower number of examples. Overall, these three metrics are much weaker factors in how well the LLM performs compared to the CFG tree depth.

## J  FEW-SHOT PROMPTING RESULTS

In this section, we discuss our few-shot prompting experiment and analyze the performance difference between zero-shot and few-shot prompting on §A1 and §A2.

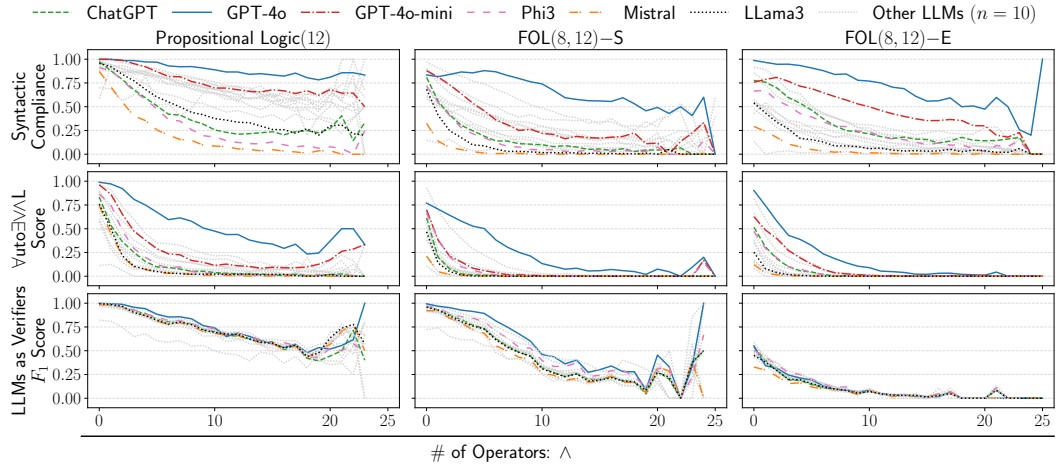

Figure 12: Zero-shot Pass@1 results (avg. over 10 batches, higher values better) from using ∀uto∃∨∧L to assess LLMs w.r.t. §A1, §A2, §A3 (Sec. 3.3.1) on the packaged datasets. The x-axis is the number of and operators (∧) in the expression.

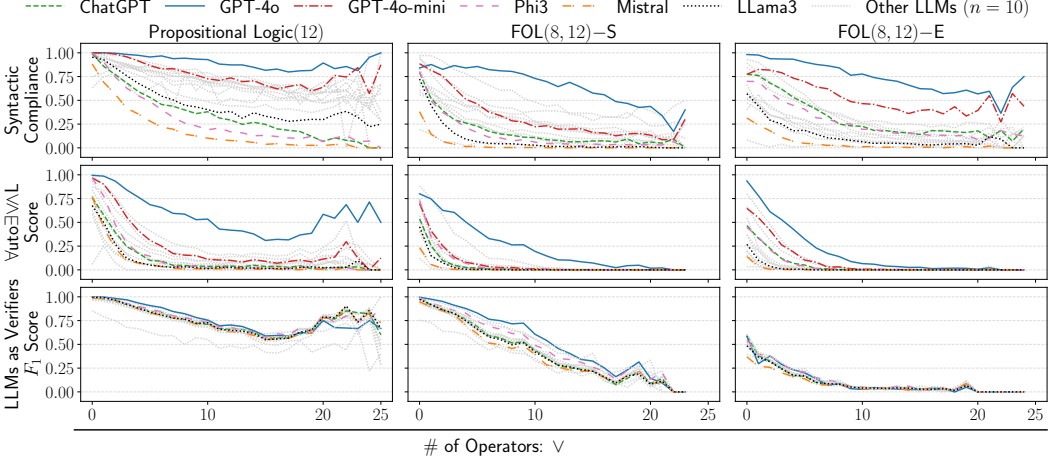

Figure 13: Zero-shot Pass@1 results (avg. over 10 batches, higher values better) from using ∀uto∃∨∧L to assess LLMs w.r.t. §A1, §A2, §A3 (Sec. 3.3.1) on the packaged datasets. The x-axis is the number of or operators (∨) in the expression.

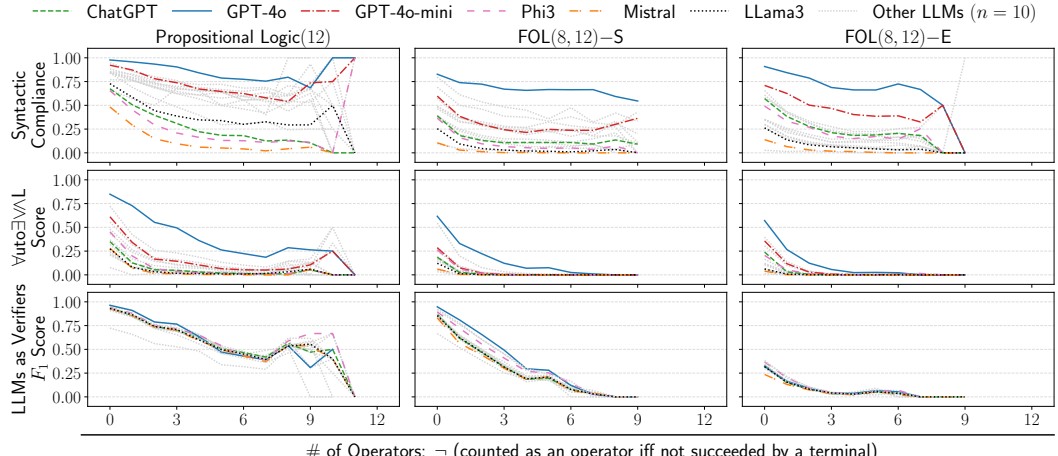

Figure 14: Zero-shot Pass@1 results (avg. over 10 batches, higher values better) from using ∀uto∃∨∧L to assess LLMs w.r.t. §A1, §A2, §A3 (Sec. 3.3.1) on the packaged datasets. The x-axis is the number of negation operators (¬) in the expression.

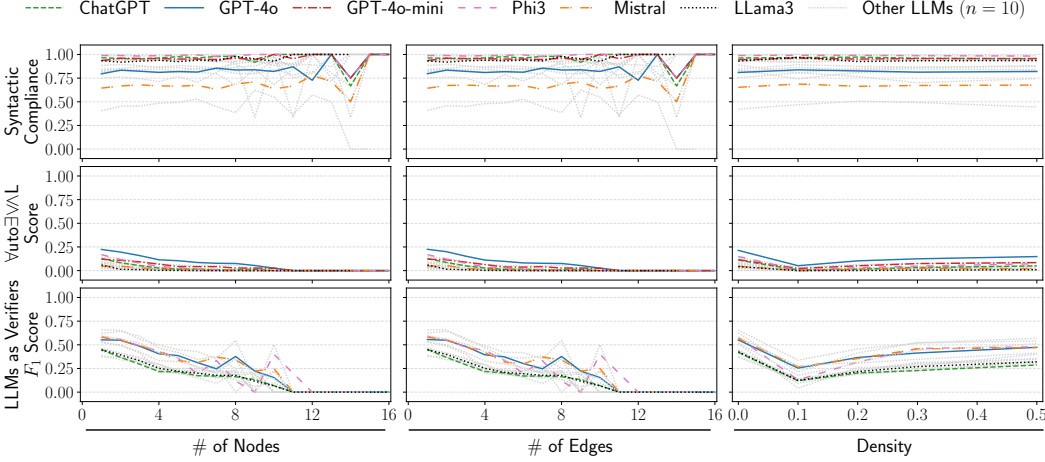

Figure 15: Zero-shot Pass@1 results (avg. over 10 batches, higher values better) from using ∀uto∃∨∧L to assess LLMs w.r.t. §A1, §A2, §A3 (Sec. 3.3.1) on the packaged datasets. The x-axis is the metric on the CFG tree to produce the regular expression formula.

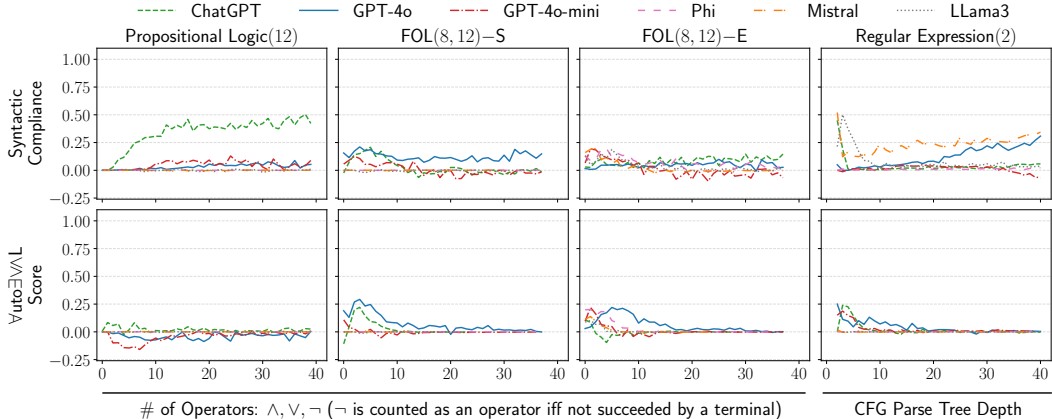

Figure 16: Syntactic compliance and accuracy difference of few-shot Pass@1 compared to zero-shot Pass@1 results (avg. over 10 batches, higher values better) from using ∀uto∃∨∧L to assess LLMs w.r.t. §A1, §A2, §A3 (Sec. 3.3.1) on the packaged datasets. The x-axis represents the increasing order of descriptional complexity.

We evaluated on the same five datasets from the main paper's experiments but inserted two examples into the prompts. First-order and predicate logic used the same two examples, while regular expressions used their own two examples. In Figure 16, the performance difference of each LLM when using few-shot prompting instead of zero-shot is shown. Using few-shot prompting increases syntactic compliance as the model has access to the desired format for encoding and decoding. For expressions with lower complexity, this translates to a better performance on §A2. However, as complexity increases, the performance difference between zero-shot and few-shot prompting is negligible due to having the correct format for parsing but failing maintaining the same formula.

# K    OTHER BENCHMARK CORRELATION AND ∀UTO∃∨∧L PREDICTIVE POWER EVALUATION

For evaluating the correlation between a LLM's performance on ∀uto∃∨∧L and existing benchmarks and measuring the predictive power of ∀uto∃∨∧L, in Section 4.2, we evaluated on FOLIO (Han et al., 2024), Multi-LogicEval (Patel et al., 2024), and HumanEval (Chen et al., 2021). In this section we discuss these experiments and cite the sources of the HumanEval results along with evaluate the predictive power of ∀uto∃∨∧L.

In this section, we discuss the experimental setup for the benchmark, the sources used for LLM performance on other benchmarks, and the ∀uto∃∨∧L we used for evaluation. We also evaluate the FOLIO premise benchmark further based on the operator numbers in each premise.

## K.1    FOLIO EXPERIMENTAL SETUPS

The FOLIO dataset is composed of premises and a conclusion for each sample where the task is to conclude whether the conclusion is true, false, or unknown given the premises. Additionally, the dataset provides an encoding into first-order logic for all the premises and conclusions. Therefore, we evaluated each LLM on their abilities to (1) informalize a first-order logic premise, (2) autoformalize a natural language premise, (3) correctly classifying the conclusion using the first-order logic representations, and (4) correctly classifying the conclusion using the natural language representations.

For the FOLIO premise informalization and autoformalization experiments, the LLM was prompted using the same few-shot first-order logic prompt used by ∀uto∃∨∧L where the example from the prompt is another premise from the same FOLIO example to make sure both the example and the evaluated premises have the same context. Premises were screened to make sure that we were able to

---

**Prompt 9: FOLIO Premise Informalization Prompt**

[TASK]
Your task is to convert a first-order logic formula, appearing after [FORMULA], to a natural description that represents the formula. Only natural language terms are allowed to be used and do not copy the formula in your description. Your description should allow one to reconstruct the formula without having access to it, so make sure to use the correct names in your description. Explicitly describe the predicates. You may use terms verbatim as specified in the vocabulary below.

[EXAMPLE 1]
$\forall x (DrinkRegularly(x, coffee) \lor (\neg WantToBeAddictedTo(x, caffeine)))$
People regularly drink coffee, or they don't want to be addicted to caffeine, or both.
[VOCABULARY]
$\lor$ represents disjunction
$\land$ represents conjunction
$\neg$ represents negation
$\rightarrow$ represents implication
( and ) represent parentheses
propositions can be used verbatim
predicates can be used verbatim
$\forall <x1><x2>...<xn>$ . represents universal quantification with x1... representing free variables
$\exists <x1><x2>...<xn>$ . represents existential quantification with x1... representing free variables
The objects are: caffeine
The parameterized predicates are: $awarethatdrug(?p0, ?p1)$, $wanttobeaddictedto(?p0, ?p1)$
The free variables are: $x$

[FORMULA]
$\forall x. (\neg wanttobeaddictedto(x, caffeine) \rightarrow \neg awarethatdrug(x, caffeine))$

---

parse them into Prover9. Prompt 9 and 10 shows example prompts using example premises come from the FOLIO dataset.

For evaluating the performance of each LLM on classifying whether the premises entailed the conclusion, the same prompt was used for both the natural language and first-order logic representations of the premises and conclusions. The prompts are inspired by the prompts used in Multi-LogiEval and use Chain-of-Thought prompting and prompt the model to provide the answer in a parsable format. Prompt 11 and Prompt 12 are examples of these prompts using an example from the FOLIO dataset.

We evaluated the informalization results against the ground truth natural language representation using BLEU (Callison-Burch et al., 2006), ROUGE (Lin, 2004), METEOR (Banerjee & Lavie, 2005), and BERT Score (Zhang* et al., 2020). The model deberta-xlarge-mnli (He et al., 2021) was used for the BERT score calculation. For the autoformalization results, we used the same verification process as the main paper. For the FOLIO conclusion classification, the LLM's answered was parsed out of its response with the examples that could not be parsed being classified as "Unknown" and marked as wrong. These examples were checked to verify the parser.

### K.2 MULTI-LOGIEVAL EXPERIMENT SETUP

The task in Multi-LogicEval (Patel et al., 2024) is to answer a yes-or-no question using the provided context, where the question was created using a certain depth of rules of logical reasoning. We used a prompt similar to the one they used where we use Chain-of-Thought prompting and prompt the LLM to provide the answer in a specific location to parse. Prompt 13 shows an example of this prompt using examples from the Multi-LogiEval dataset.

---

**Prompt 10: FOLIO Premise Autoformalization Prompt**

[VOCABULARY]
Use $\vee$ to represent disjunction
Use $\wedge$ to represent conjunction
Use $\neg$ to represent negation
Use ( and ) to represent parentheses
The objects are: caffeine
The parameterized predicates are: $awarethatdrug(?p0, ?p1)$, $wanttobeaddictedto(?p0, ?p1)$
The free variables are: $x$

[TASK]
Your task is to interpret the natural language (NL) description of a first-order logic formula and represent it as formal syntax using the vocabulary specified in the [VOCABULARY] block above. Only output the formula and no other text. The NL description appears immediately following the [NL DESCRIPTION] tag.

[EXAMPLE 1]
People regularly drink coffee, or they don't want to be addicted to caffeine, or both.
$\forall x(DrinkRegularly(x, coffee) \vee (\neg WantToBeAddictedTo(x, caffeine)))$

[NL DESCRIPTION]
No one who doesn't want to be addicted to caffeine is unaware that caffeine is a drug.

---

**Prompt 11: FOLIO Natural Language Representation Prompt**

For the following [PREMISES] containing rules of logical reasoning, perform step-by-step reasoning to answer whether the [CONCLUSION] is True/False/Uncertain based on the [PREMISES]. Use the following answer format:
Reasoning Steps:
Answer: True/False/Uncertain
[PREMISES]:
All people who regularly drink coffee are dependent on caffeine
People regularly drink coffee, or they don't want to be addicted to caffeine, or both.
No one who doesn't want to be addicted to caffeine is unaware that caffeine is a drug.
Rina is either a student who is unaware that caffeine is a drug, or she is not a student and is she aware that caffeine is a drug.
Rina is either a student who depend on caffeine, or she is not a student and not dependent on caffeine.
[CONCLUSION]:
Rina doesn't want to be addicted to caffeine or is unaware that caffeine is a drug.

---

### K.3 HUMANEVAL AND BIG BENCH HARD SCORE SOURCES

To evaluate the correlation and predictive power of $\forall$uto$\exists\vee\wedge$L against commonly used LLM benchmarks HumanEval (Chen et al., 2021) and Big Bench Hard (BBH) (Suzgun et al., 2023), we collected the performance scores of the LLMs we evaulated on both benchmarks and report our findings and sources in Table 6. We were unable to find any sources that evaluated GPT-4o-mini on BBH.

### K.4 COMPUTED CALIBRATED $\forall$UTO$\exists\vee\wedge$L SCORE

To compare against the performance on different benchmarks in Section 4.2, we needed to calculate the calibrated performance of each LLM on $\forall$uto$\exists\vee\wedge$L for the relevant portions of the datasets. For example, there are few premises in the FOLIO dataset with more than 6 operators meaning that the most accurate comparison would be to evaluate our first-order logic dataset up to the same number of

---

**Prompt 12: FOLIO First-Order Logic Representation Prompt**

For the following [PREMISES] containing rules of logical reasoning, perform step-by-step reasoning to answer whether the [CONCLUSION] is True/False/Uncertain based on the [PREMISES]. Use the following answer format:
Reasoning Steps:
Answer: True/False/Uncertain
[PREMISES]:
$\forall x(DrinkRegularly(x, coffee) \rightarrow IsDependentOn(x, caffeine))$
$\forall x(DrinkRegularly(x, coffee) \vee (\neg WantToBeAddictedTo(x, caffeine)))$
$\forall x(\neg WantToBeAddictedTo(x, caffeine) \rightarrow \neg AwareThatDrug(x, caffeine))$
$\neg(Student(rina) \oplus \neg AwareThatDrug(rina, caffeine))$
$\neg(IsDependentOn(rina, caffeine) \oplus Student(rina))$
[CONCLUSION]:
$\neg WantToBeAddictedTo(rina, caffeine) \vee (\neg AwareThatDrug(rina, caffeine))$

---

**Prompt 13: Multi-LogicEval Prompt**

"Given the context that contains rules of logical reasoning in natural language and question, perform step-by-step reasoning to answer the question. Based on context and reasoning steps, answer the question ONLY in 'yes' or 'no.' Please use the below format:
Context: At a university, students who study hard earn high grades. Those who participate in extracurriculars develop leadership skills. However, students have restricted time outside of classes. They can either study hard or they do not develop leadership skills from extracurriculars.
Question: Can we conclude that Priya, a university student with limited free time, either earns high grades or does not participate in extracurricular activities?
Reasoning steps: [generate step-by-step reasoning]
Answer: Yes/No"

---

Table 6: Reported performance of SOTA LLMs on HumanEval and Big Bench Hard (BBH) benchmarks. The values under the Computed column are averaged over 5 runs from our experiments. Other results are reported from online sources. A – indicates that we were not able to find any online source. We used our local computed results when they were available.

| Model | HumanEval Score | | BBH Score |
|---|---|---|---|
| | **Computed** | **(Online)** | **(Online)** |
| ChatGPT | 74.3 | 68 (OpenAI, 2024) | 48.1 (OpenAI, 2023) |
| GPT-4o | 91.8 | 90.2 (OpenAI, 2024) | 83.1 (Dunham & Syahputra, 2024) |
| GPT-4o-mini | 88.3 | 87.2 (OpenAI, 2024) | – |
| Llama-3.2-1B-Instruct | 34.6 | – | 8.7 (Fourrier et al., 2024) |
| Qwen-2.5-1.5B-Instruct | 56.7 | 61.6 (Qwen2, 2024) | 19.8 (Fourrier et al., 2024) |
| Phi-3.5-Mini-Instruct | 71.3 | 64.6 (Liu et al., 2023) | 36.7 (Fourrier et al., 2024) |
| Mistral-7B-Instruct-v0.2 | 44.5 | 42.1 (Liu et al., 2023) | 24.0 (Fourrier et al., 2024) |
| Llama-3-8B-Instruct | 62.8 | 61.6 (Liu et al., 2023) | 24.5 (Fourrier et al., 2024) |
| Granite-3.0-8B-Instruct | 62.2 | 64.6 (Granite Team, 2024) | 51.6 (Fourrier et al., 2024) |
| Llama-3.1-8B-Instruct | 63.4 | 66.5 (Microsoft, 2024) | 63.4 (Microsoft, 2024) |
| Ministral-8B-Instruct-2410 | 76.8 | 76.8 (MistralAI, 2024) | 8.7 (Fourrier et al., 2024) |
| Gemma-2-9B-IT | 68.3 | 68.9 (Qwen2, 2024) | 42.1 (Fourrier et al., 2024) |
| Phi-3-Medium-4k-Instruct | 75.0 | 62.2 (Microsoft, 2024) | 49.4 (Fourrier et al., 2024) |
| Qwen-2.5-14B-Instruct | 80.5 | 83.5 (Qwen2, 2024) | 48.4 (Fourrier et al., 2024) |
| Yi-1.5-34B-Instruct | 72.6 | 75.2 (Yi, 2024) | 44.3 (Fourrier et al., 2024) |
| Llama-3-70B-Instruct | 79.9 | 77.4 (Liu et al., 2023) | 50.2 (Fourrier et al., 2024) |

operators. Therefore, we calculated the accuracy of the first-order logic formulae with less than seven operators when calculating the correlation and predictive power. On MultiLogiEval, the number of

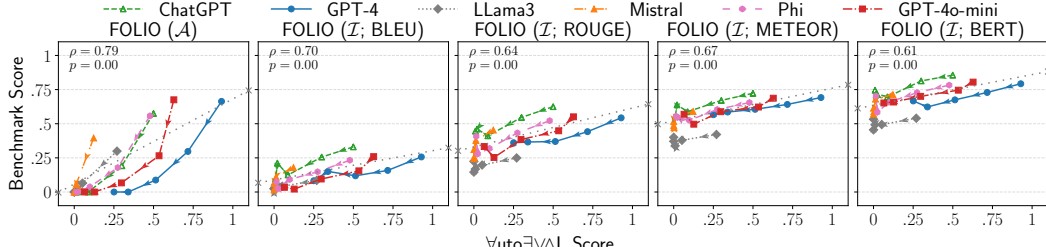

Figure 17: Correlation between the parameterized ∀uto∃∨∧L score and both autoformalization $\mathcal{A}$ and informalization $\mathcal{I}$ for FOLIO premises. Each point represents a specific number of operators with arrows showing increasing complexity (number of operators). The trendline across all the points is annotated with ×, the Pearson correlation coefficient ($\rho$), and the p-value are annotated in the top left.

operators is dictated by the depth of the rules, so we took the average of all first-order logic examples up to 30 in our dataset. On HumanEval, to the best of our knowledge using the average of regex with CFG tree depth up to 7 is the best comparison.

### K.5 FOLIO ADDITIONAL CORRELATION FIGURES

In Section 4.2, we evaluated the correlation of other benchmarks compared to ∀uto∃∨∧L. For the FOLIO dataset, we were able to calculate the exact number of operators in each problem, allowing us to plot points comparing the autoformalization and informalization accuracy for each operator number class to directly compare to the accuracy of the same number of operators in the first-order logic dataset we generated.

We plot these results in Figure 17 with the Pearson correlation coefficient. Each figure shows a moderate to strong correlation with a statistically significant p-value of less than 0.05. As the computational complexity increases, performance on ∀uto∃∨∧L, autoformalization, and informalization decreases. The autoformalization correlation is significantly stronger due to the informalization evaluation metrics being much weaker at evaluating truth maintenance.

## L LLM AS VERIFIERS EVALUATION

In this section, we analyze the performance of LLMs on §A3, where we evaluate the performance of using a LLM to verify whether the formal syntax expression $\varphi$ is equivalent to the one produced by GPT-4o after doing $\mathcal{A} \circ \mathcal{I}(\varphi)$. Figures 21 and 22 show the number of positive and negative examples. Additionally, it breaks down the performance of each LLM on each dataset.

The LLMs are clearly biased towards giving positive answers with a high ratio of false positives to negative labels compared to false negatives to positive labels. A common case was the LLM not recognizing that GPT-4o renamed predicates or constants producing a different formal syntax expression. A structure that ChatGPT tends to struggle with is double negations. Below we provide two examples where, even at the time of writing this rebuttal, ChatGPT failed to correctly verify whether the two formulae are equivalent. Note that [FORMULA 2] is ChatGPT's own response after conducting $(\mathcal{A} \circ \mathcal{I})^n(\varphi_0)$ where $\varphi_0 =$ [FORMULA 1].

In the case of Output Example 1, it can be clearly seen that ChatGPT failed to understand that the two formulae are different. In particular, it was not able to correctly evaluate or reason that the second regular expression represents a string that begins with a 1 followed by zero or one 0's. This is despite the LLM using a "Chain-of-thought" to help it arrive at an answer.

A more glaring reasoning error is found in Output Example 2, where ChatGPT fails to apply the principles of logic where $\neg\neg p \equiv p \not\equiv \neg p$. Our results convincingly showcase that LLMs cannot be used as verifiers even for straightforward statements such as the examples presented.

---

**Output Example 1: Regular Expression Verification Example**

*Prompt:*
Your task is to say whether two Regular Expression formulae are equivalent. The first formula will appear right after [FORMULA 1] and the second after [FORMULA 2].
Give an explanation followed by a yes or no answer. The answer must show up at the end with the format "[Answer]" followed by either a yes or no.
[FORMULA 1]: 100*
[FORMULA 2]: 1(0*)

- - - - - - - - - - - - - - - - - - - - - - - - - - - - - - - - - - - - - - - - -

*Response:*
To determine whether the two regular expressions $100*$ and $1(0*)$ are equivalent, let's analyze them:
### Explanation:
1. **Formula 1: 100\***
- This matches the character "1" followed by zero or more "0"s.
- It will match: "1", "10", "100", "1000", etc.
2.**Formula 2: 1(0\*)**
- This matches the character "1" followed by zero or more "0"s (the same as Formula 1).
- It will also match: "1", "10", "100", "1000", etc.
### Key Observation:
- Both formulas describe the same set of strings: a "1" followed by zero or more "0"s.
- The use of parentheses in Formula 2 is purely for grouping and does not alter the matching behavior.
### Conclusion:
The two regular expressions are equivalent.
**[Answer]: yes**

---

**Output Example 2: First-Order Logic Verification Example**

*Prompt:*
Your task is to say whether two First Order Logic formulae are equivalent. The first formula will appear right after [FORMULA 1] and the second after [FORMULA 2].
Give an explanation followed by a yes or no answer. The answer must show up at the end with the format "[Answer]" followed by either a yes or no.
[FORMULA 1]: $\forall x1.(\neg\neg pred8(p8, p7) \vee \neg pred4(x1))$
[FORMULA 2]: $\forall x1.(\neg pred8(p8, p7) \vee \neg pred4(x1))$

- - - - - - - - - - - - - - - - - - - - - - - - - - - - - - - - - - - - - - - - -

*Response:*
The two formulas are equivalent.
Explanation: In both Formula 1 and Formula 2, the part ¬¬pred8(p8, p7) is equivalent to ¬pred8(p8, p7) due to the double negation elimination rule in logic. Therefore, both formulas convey the same meaning, as the presence or absence of a double negation does not affect the truth value in this context.
[Answer]: Yes

---

## M  DATASET DIVERSITY

Fig. 9 and Fig. 9 provide additional details on the types of data present in the datasets packaged with $\forall$uto$\exists\vee\wedge$L. Users can generate dynamic datasets along these dimensions using the hyperparameters mentioned in Table 1.

To further provide additional statistics pertaining to the similarity of formulae in our dataset, especially those where the formulae are otherwise equivalent but just use different vocabularies. For example, the formula $f = p_1$ can be represented via different propositions where $p_1 = $ *It is raining* in $f_1$ and

something different in another formula $f_2$ even though they canonically represent the same formula $f$.

This allows to test robustness in LLM outputs. Nevertheless, the probability of such instances decreases as the formula size increases. We have counted the total proporition of the dataset where this occurs by replacing any variable from the vocabulary with an element of a vocabulary of size 1. For example, all variables used in PL(12) dataset of our results are replaced by substituting those variables with a vocabulary of only 1 proposition. Excess parentheses etc are preprocessed using NLTK and removed before the substitution (e.g. $((p_1) \wedge p_2)$ is simplified to $p_1 \wedge p_2$.

The k-CNF dataset contains 8550 unique samples and the propositional logic dataset contains 17.7k samples constituting 85% and 90% of these datasets respectively.

# N    EVALUATION OF LLMS

Table 7 lists the models, their parameters (– if closed-source), and the exact model version used for our experiments. The open-source models were loaded using NVIDIA A100 80GB GPUs whereas we used the OpenAI API for the GPT family of models. We cover a diverse range of models in our evaluation ranging from extremely small LMs with a few billion parameters ($\sim$1B) to LLMs with several billions of parameters. This allows the analysis of $\forall$uto$\exists\vee\wedge$L from the lens of generalization.

Fig. 18 represents the syntactic compliance (§A1) data from Fig. 3 for all the models with a separate axis for each LLM. Similarly, Fig. 19 plots the Accuracy (§A2). Additionally, Fig. 20 plots the F1 score of using LLMs as verifiers (§A3). Tables 9 – 14 provide the data that was used to plot the results in Fig. 4 and to compute the predictive power in Fig. 5.

Tables 15 – 18 list the example counts for each combination of class label and prediction for FOLIO(R; NL) and FOLIO(R; FOL) and each label's precision and recall rate. Tables 19 and 20 list the examples counts for each combination of class label and prediction for LogiEval(R; PL) and LogieEval(R; FOL).

## N.1    CLAUDE EVALUATION

We evaluated Claude 3.0 Sonnet on just the 3-CNF, propositional logic, and regular expression datasets due to the cost. Our results are shown in Figure 23 and show that Claude 3.0 Sonnet performs similarly to GPT-4o with both having nearly perfect syntactic compliance and accuracy on 3-CNF. Sonnet achieved the highest syntactic compliance and accuracy on propositional logic compared to the other models. However the accuracy was only around 50% for expressions with more than 20 operators. Additionally, while being often syntactic compliant, Sonnet performed with low accuracy on the regular expression dataset.

Table 7: The LLMs used in our evaluation. The label names represent the labels used in Fig. 18 and Fig. 19, $|\theta|$ represents the total number of parameters, and the last column lists the exact version used (for reproducibility).

| Label | $|\theta|$ | Version |
|---|---|---|
| ChatGPT | – | `GPT-3.5-turbo-0125` |
| GPT-4o | – | `gpt-4o-2024-08-06` |
| GPT-4o-mini | – | `gpt-4o-mini-2024-07-18` |
| GPT-4o1 | – | `o1-preview-2024-09-12` |
| Llama-3.2-1B-Instruct | 1B | `meta-llama/Llama-3.2-1B-Instruct` |
| Qwen-2.5-1.5B-Instruct | 1.5B | `Qwen/Qwen2.5-1.5B-Instruct` |
| Phi-3.5-Mini-Instruct | 4B | `microsoft/Phi-3.5-mini-instruct` |
| Mistral-7B-Instruct-v0.2 | 7B | `mistralai/Mistral-7B-Instruct-v0.2` |
| Llama-3-8B-Instruct | 8B | `meta-llama/Llama-3-8B-Instruct` |
| Granite-3.0-8B-Instruct | 8B | `ibm-granite/granite-3.0-8b-instruct` |
| LLama-3.1-8B-Instruct | 8B | `meta-llama/Llama-3.1-8B-Instruct` |
| Ministral-8B-Instruct-2410 | 8B | `mistralai/Ministral-8B-Instruct-2410` |
| Gemma-2-9B-IT | 9B | `google/gemma-2-9b-it` |
| Phi-3-Medium-4k-Instruct | 14B | `microsoft/Phi-3-medium-4k-instruct` |
| Qwen-2.5-14B-Instruct | 14B | `Qwen/Qwen2.5-14B-Instruct` |
| Yi-1.5-34B-Instruct | 34B | `01-ai/Yi-34B-Instruct` |
| Llama-3-70B-Instruct | 70B | `meta-llama/Llama-3-70B-Instruct` |

Table 8: Correlation data for FOLIO(R; NL). The $\forall$uto$\exists\lor\land$L data was averaged from the PL dataset with data points with description complexity $d \leq 6$. These values were used to compute the predictive power of $\forall$uto$\exists\lor\land$L reported in Fig. 5.

| Model | $\forall$uto$\exists\lor\land$L Score | FOLIO(R; NL) Score |
|---|---|---|
| GPT-4o | 0.79 | 0.75 |
| GPT-4o-mini | 0.56 | 0.69 |
| ChatGPT | 0.36 | 0.56 |
| Mistral-7B-Instruct-v0.2 | 0.06 | 0.54 |
| Phi-3-medium-4k-instruct | 0.35 | 0.67 |
| LLama-3-8B-Instruct | 0.13 | 0.58 |
| Gemma-2-9B-IT | 0.28 | 0.64 |
| Granite-3.0-8B-Instruct | 0.18 | 0.60 |
| Llama-3.1-8B-Instruct | 0.09 | 0.59 |
| LLama-3.2-1B-Instruct | 0.03 | 0.36 |
| LLama-3-70B-Instruct | 0.49 | 0.70 |
| Ministral-8B-Instruct-2410 | 0.10 | 0.61 |
| Phi-3.5-Mini-Instruct | 0.19 | 0.61 |
| Qwen-2.5-1.5B-Instruct | 0.07 | 0.49 |
| Qwen-2.5-14B-Instruct | 0.67 | 0.73 |
| Yi-1.5-34B | 0.21 | 0.63 |

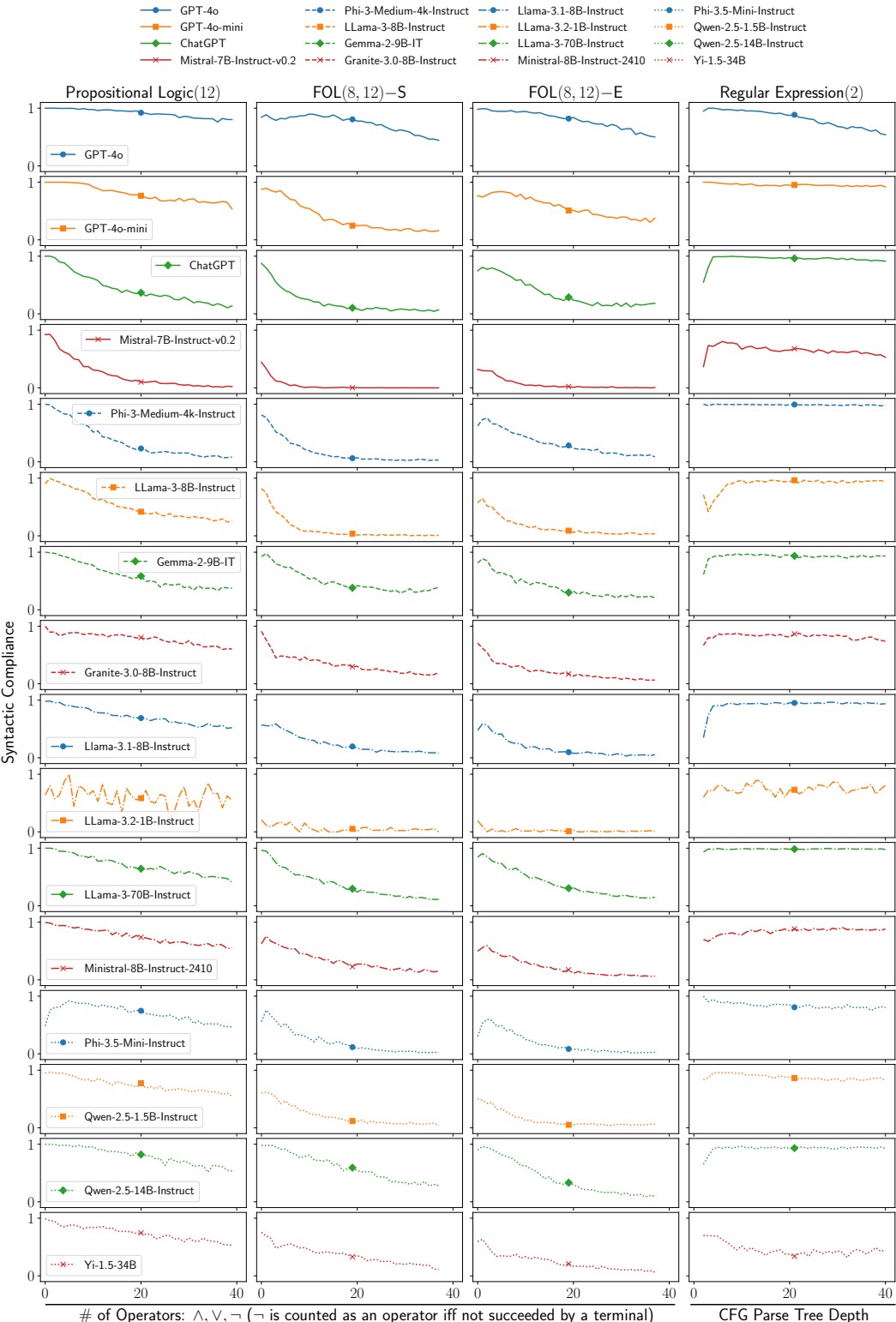

Figure 18: Syntactic compliance (§A1) of all models on the ∀uto∃∨∧L datasets.

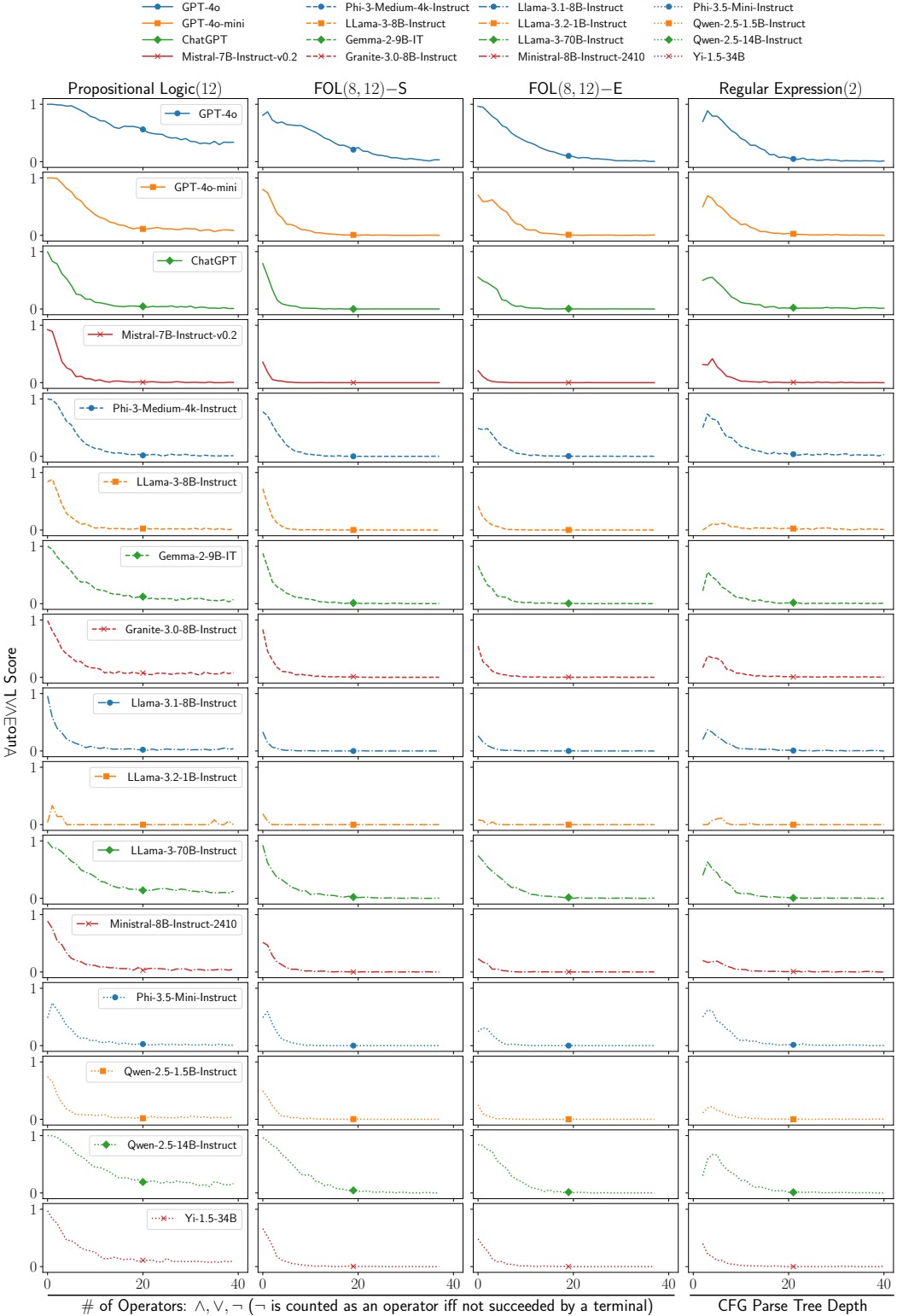

Figure 19: ∀uto∃∨∧L Score (§A2) of all models on the ∀uto∃∨∧L datasets.

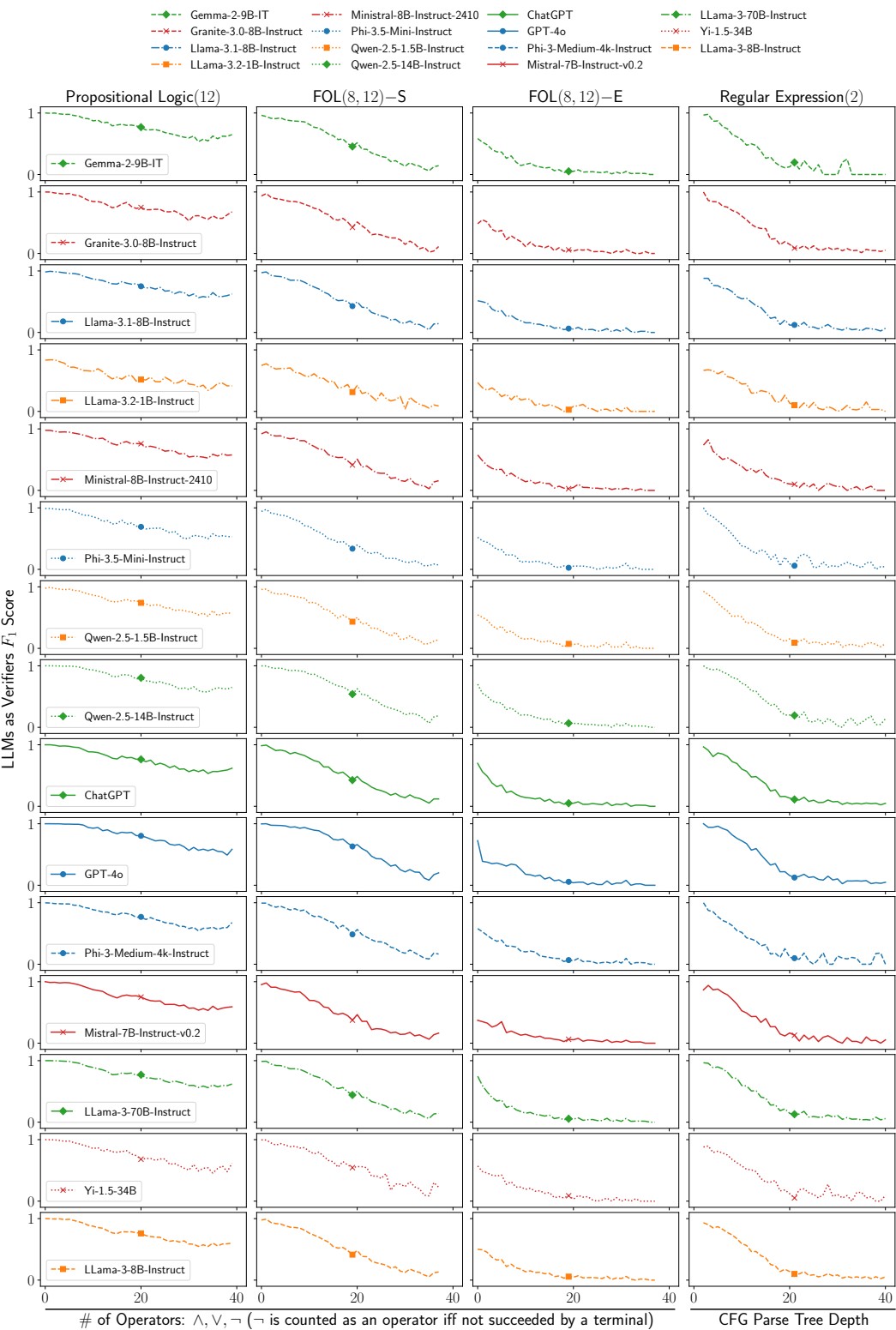

Figure 20: LLMs as verifiers F1 score (§A3) for the GPT-4o results using the ∀uto∃∨∧L procedure of all models on the ∀uto∃∨∧L datasets.

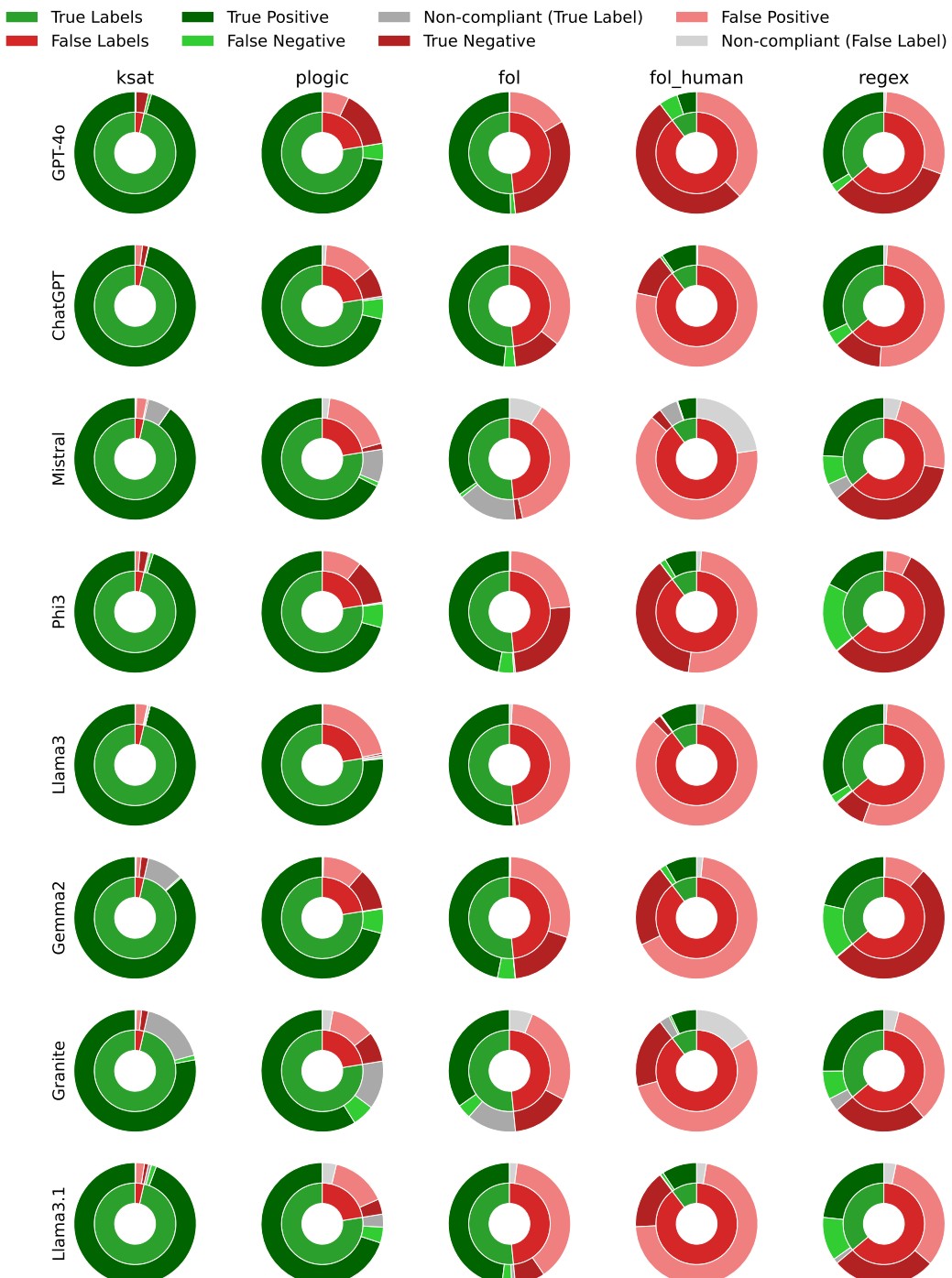

Figure 21: The number of positive and negative examples of $\varphi_1 \equiv \mathcal{A} \circ \mathcal{I}(\varphi_0)$ when evaluating GPT-4o on $\forall$uto$\exists\lor\land$L for each dataset (inner donuts). Additionally included is a breakdown of the performance of each LLM when acting as the verifier (outer donuts). Included are all examples containing 20 or fewer operators or, in the case of the regular expression dataset, CFG tree depth of 20 or fewer. Non-compliant represents syntactically non-compliant responses when prompted to verify the equivalence.

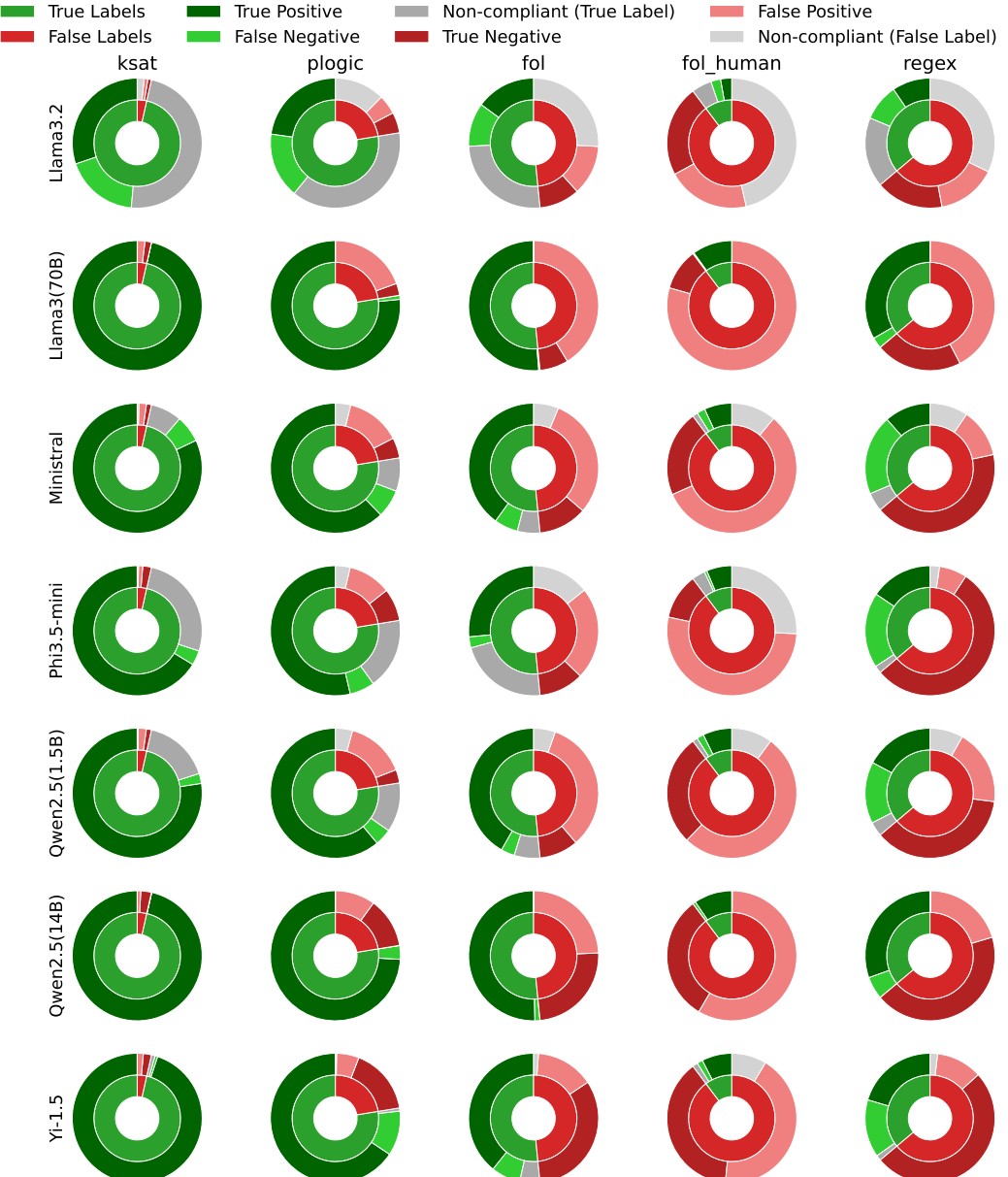

Figure 22: Additional LLMs evaluate on §A3, where the number of positive and negative examples of $\varphi_1 \equiv \mathcal{A} \circ \mathcal{I}(\varphi_0)$ when evaluating GPT-4o on $\forall$uto$\exists\lor\land$L for each dataset (inner donuts). Additionally included is a breakdown of the performance of each LLM when acting as the verifier (outer donuts). Included are all examples containing 20 or fewer operators or, in the case of the regular expression dataset, CFG tree depth of 20 or fewer. Non-compliant represents syntactically non-compliant responses when prompted to verify the equivalence.

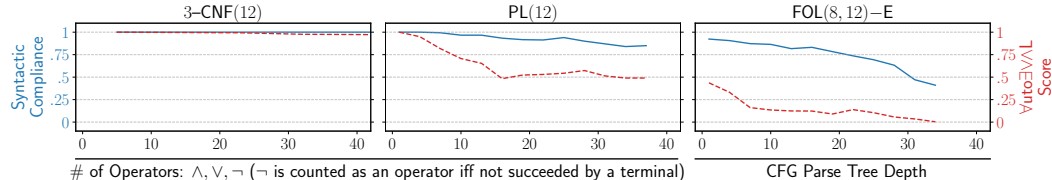

Figure 23: ∀uto∃∨∧L results on Claude 3.0 Sonnet on the 3-CNF, propositional logic, and regular expression datasets. Dashed line is the accuracy.

Table 9: Correlation data for FOLIO(R; FOL). The calibrated ∀uto∃∨∧L score was calculated from the FOL dataset with data points with description complexity $d \leq 6$. These values were used to compute the predictive power of ∀uto∃∨∧L reported in Fig. 5.

| Model | ∀uto∃∨∧L Score | FOLIO(R; FOL) Score |
|---|---|---|
| GPT-4o | 0.79 | 0.71 |
| GPT-4o-mini | 0.56 | 0.67 |
| ChatGPT | 0.36 | 0.51 |
| Mistral-7B-Instruct-v0.2 | 0.06 | 0.51 |
| Phi-3-medium-4k-instruct | 0.35 | 0.62 |
| LLama-3-8B-Instruct | 0.13 | 0.52 |
| Gemma-2-9B-IT | 0.28 | 0.59 |
| Granite-3.0-8B-Instruct | 0.18 | 0.56 |
| Llama-3.1-8B-Instruct | 0.09 | 0.56 |
| LLama-3.2-1B-Instruct | 0.03 | 0.36 |
| LLama-3-70B-Instruct | 0.49 | 0.66 |
| Ministral-8B-Instruct-2410 | 0.10 | 0.56 |
| Phi-3.5-Mini-Instruct | 0.19 | 0.53 |
| Qwen-2.5-1.5B-Instruct | 0.07 | 0.45 |
| Qwen-2.5-14B-Instruct | 0.67 | 0.71 |
| Yi-1.5-34B | 0.21 | 0.61 |

Table 10: Correlation data for LogiEval(R; PL). The calibrated ∀uto∃∨∧L score was calculated from the PL dataset with data points with description complexity $d \leq 30$. These values were used to compute the predictive power of ∀uto∃∨∧L reported in Fig. 5.

| Model | ∀uto∃∨∧L Score | LogiEval(R; PL) Score |
|---|---|---|
| GPT-4o | 0.67 | 0.87 |
| GPT-4o-mini | 0.35 | 0.67 |
| ChatGPT | 0.17 | 0.64 |
| Mistral-7B-Instruct-v0.2 | 0.12 | 0.60 |
| Phi-3-medium-4k-instruct | 0.23 | 0.75 |
| LLama-3-8B-Instruct | 0.12 | 0.61 |
| Gemma-2-9B-IT | 0.28 | 0.71 |
| Granite-3.0-8B-Instruct | 0.21 | 0.58 |
| Llama-3.1-8B-Instruct | 0.11 | 0.71 |
| LLama-3.2-1B-Instruct | 0.04 | 0.50 |
| LLama-3-70B-Instruct | 0.34 | 0.85 |
| Ministral-8B-Instruct-2410 | 0.17 | 0.68 |
| Phi-3.5-Mini-Instruct | 0.10 | 0.62 |
| Qwen-2.5-1.5B-Instruct | 0.11 | 0.52 |
| Qwen-2.5-14B-Instruct | 0.46 | 0.76 |
| Yi-1.5-34B | 0.26 | 0.78 |

Table 11: Correlation data for LogiEval(R; FOL). The calibrated ∀uto∃∨∧L score was calculated from the FOL dataset with data points with description complexity $d \leq 30$. These values were used to compute the predictive power of ∀uto∃∨∧L reported in Fig. 5.

| Model | ∀uto∃∨∧L Score | LogiEval(R; FOL) Score |
|---|---|---|
| GPT-4o | 0.32 | 0.82 |
| GPT-4o-mini | 0.17 | 0.56 |
| ChatGPT | 0.09 | 0.63 |
| Mistral-7B-Instruct-v0.2 | 0.01 | 0.56 |
| Phi-3-medium-4k-instruct | 0.09 | 0.70 |
| LLama-3-8B-Instruct | 0.02 | 0.62 |
| Gemma-2-9B-IT | 0.07 | 0.69 |
| Granite-3.0-8B-Instruct | 0.05 | 0.55 |
| Llama-3.1-8B-Instruct | 0.02 | 0.68 |
| LLama-3.2-1B-Instruct | 0.00 | 0.47 |
| LLama-3-70B-Instruct | 0.15 | 0.78 |
| Ministral-8B-Instruct-2410 | 0.02 | 0.64 |
| Phi-3.5-Mini-Instruct | 0.04 | 0.54 |
| Qwen-2.5-1.5B-Instruct | 0.02 | 0.50 |
| Qwen-2.5-14B-Instruct | 0.19 | 0.66 |
| Yi-1.5-34B | 0.05 | 0.71 |

Table 12: Correlation data for FOLIO($\mathcal{A}$). The calibrated ∀uto∃∨∧L score was calculated from the FOL dataset with data points with description complexity $d \leq 6$. These values were used to compute the predictive power of ∀uto∃∨∧L reported in Fig. 5.

| Model | ∀uto∃∨∧L Score | FOLIO($\mathcal{A}$) Score |
|---|---|---|
| GPT-4o | 0.79 | 0.41 |
| GPT-4o-mini | 0.56 | 0.40 |
| ChatGPT | 0.36 | 0.33 |
| Mistral-7B-Instruct-v0.2 | 0.06 | 0.19 |
| Phi-3-medium-4k-instruct | 0.35 | 0.32 |
| LLama-3-8B-Instruct | 0.13 | 0.16 |
| Gemma-2-9B-IT | 0.28 | 0.30 |
| Granite-3.0-8B-Instruct | 0.18 | 0.14 |
| Llama-3.1-8B-Instruct | 0.09 | 0.23 |
| LLama-3.2-1B-Instruct | 0.03 | 0.00 |
| LLama-3-70B-Instruct | 0.49 | 0.40 |
| Ministral-8B-Instruct-2410 | 0.10 | 0.23 |
| Phi-3.5-Mini-Instruct | 0.19 | 0.18 |
| Qwen-2.5-1.5B-Instruct | 0.07 | 0.10 |
| Qwen-2.5-14B-Instruct | 0.67 | 0.36 |
| Yi-1.5-34B | 0.21 | 0.31 |

Table 13: Correlation data for FOLIO($\mathcal{I}$). The ∀uto∃∨∧L data was averaged from the FOL dataset with data points with description complexity $d \leq 6$. These values were used to compute the predictive power of ∀uto∃∨∧L reported in Fig. 5.

| | | FOLIO($\mathcal{I}$) Score | | | |
|---|---|---|---|---|---|
| Model | ∀uto∃∨∧L Score | BLEU | ROUGE | METEOR | BERT |
| GPT-4o | 0.79 | 0.14 | 0.42 | 0.64 | 0.71 |
| GPT-4o-mini | 0.56 | 0.13 | 0.41 | 0.61 | 0.73 |
| ChatGPT | 0.36 | 0.19 | 0.47 | 0.62 | 0.76 |
| Mistral-7B-Instruct-v0.2 | 0.06 | 0.08 | 0.31 | 0.51 | 0.64 |
| Phi-3-Medium-4k-Instruct | 0.35 | 0.12 | 0.39 | 0.58 | 0.70 |
| LLama-3-8B-Instruct | 0.13 | 0.04 | 0.18 | 0.35 | 0.50 |
| Gemma-2-9B-IT | 0.28 | 0.10 | 0.34 | 0.53 | 0.63 |
| Granite-3.0-8B-Instruct | 0.18 | 0.15 | 0.41 | 0.58 | 0.72 |
| Llama-3.1-8B-Instruct | 0.09 | 0.09 | 0.31 | 0.51 | 0.64 |
| LLama-3.2-1B-Instruct | 0.03 | 0.00 | 0.06 | 0.15 | 0.36 |
| LLama-3-70B-Instruct | 0.49 | 0.12 | 0.40 | 0.60 | 0.70 |
| Ministral-8B-Instruct-2410 | 0.10 | 0.11 | 0.36 | 0.55 | 0.67 |
| Phi-3.5-Mini-Instruct | 0.19 | 0.05 | 0.22 | 0.41 | 0.55 |
| Qwen-2.5-1.5B-Instruct | 0.07 | 0.09 | 0.33 | 0.49 | 0.65 |
| Qwen-2.5-14B-Instruct | 0.67 | 0.07 | 0.26 | 0.45 | 0.55 |
| Yi-1.5-34B | 0.21 | 0.12 | 0.39 | 0.58 | 0.72 |

Table 14: Correlation data for HumanEval ($\mathcal{A}$). The ∀uto∃∨∧L data was averaged from the regex dataset with data points with description complexity $d \leq 7$. These values were used to compute the predictive power of ∀uto∃∨∧L reported in Fig. 5.

| Model | ∀uto∃∨∧L Score | HumanEval ($\mathcal{A}$) Score |
|---|---|---|
| GPT-4o | 0.66 | 0.92 |
| GPT-4o-mini | 0.44 | 0.88 |
| ChatGPT | 0.36 | 0.74 |
| Mistral-7B-Instruct-v0.2 | 0.20 | 0.44 |
| Phi-3-medium-4k-instruct | 0.45 | 0.75 |
| LLama-3-8B-Instruct | 0.07 | 0.63 |
| Gemma-2-9B-IT | 0.28 | 0.68 |
| Granite-3.0-8B-Instruct | 0.21 | 0.62 |
| Llama-3.1-8B-Instruct | 0.19 | 0.63 |
| LLama-3.2-1B-Instruct | 0.03 | 0.35 |
| LLama-3-70B-Instruct | 0.33 | 0.80 |
| Ministral-8B-Instruct-2410 | 0.13 | 0.77 |
| Phi-3.5-Mini-Instruct | 0.36 | 0.71 |
| Qwen-2.5-1.5B-Instruct | 0.12 | 0.57 |
| Qwen-2.5-14B-Instruct | 0.45 | 0.80 |
| Yi-1.5-34B | 0.13 | 0.73 |

Table 15: Count of examples in FOLIO(R; NL) for each combination of (T)rue, (F)alse, and (U)ncertain label and predictions in that order. For example, TU is the number of times a LLM predicted a True label as Uncertain.

| Model | TT | TF | TU | FT | FF | FU | UT | UF | UU |
|---|---|---|---|---|---|---|---|---|---|
| GPT-4o | 1667 | 125 | 147 | 178 | 1133 | 131 | 246 | 419 | 952 |
| GPT-4o-mini | 1500 | 187 | 253 | 162 | 1087 | 196 | 255 | 488 | 877 |
| ChatGPT | 1501 | 281 | 147 | 366 | 889 | 186 | 620 | 562 | 434 |
| Mistral-7B-Instruct-v0.2 | 1301 | 205 | 412 | 200 | 717 | 508 | 525 | 387 | 691 |
| Phi-3-medium-4k-instruct | 1468 | 153 | 310 | 222 | 871 | 343 | 310 | 289 | 1009 |
| LLama-3-8B-Instruct | 1400 | 244 | 288 | 272 | 807 | 347 | 477 | 413 | 717 |
| Gemma-2-9B-IT | 1439 | 94 | 385 | 231 | 813 | 382 | 378 | 289 | 934 |
| Granite-3.0-8B-Instruct | 1415 | 109 | 412 | 304 | 666 | 471 | 382 | 316 | 919 |
| Llama-3.1-8B-Instruct | 1400 | 174 | 314 | 267 | 766 | 360 | 435 | 352 | 781 |
| LLama-3.2-1B-Instruct | 1407 | 235 | 118 | 931 | 279 | 81 | 1109 | 235 | 111 |
| LLama-3-70B-Instruct | 1677 | 101 | 161 | 263 | 966 | 214 | 419 | 319 | 881 |
| Ministral-8B-Instruct-2410 | 1577 | 242 | 116 | 358 | 954 | 130 | 583 | 532 | 503 |
| Phi-3.5-Mini-Instruct | 1355 | 164 | 398 | 223 | 821 | 383 | 357 | 359 | 890 |
| Qwen-2.5-1.5B-Instruct | 1470 | 345 | 117 | 615 | 684 | 138 | 824 | 477 | 309 |
| Qwen-2.5-14B-Instruct | 1564 | 132 | 239 | 215 | 1020 | 207 | 266 | 276 | 1058 |
| Yi-1.5-34B | 1507 | 125 | 298 | 240 | 894 | 305 | 497 | 346 | 773 |

Table 16: Calculated precision and recall for each label in FOLIO (R;NL).

| | True Label | | False Label | | Uncertain Label | |
|---|---|---|---|---|---|---|
| Model | Prec. | Rec. | Prec. | Rec. | Prec. | Rec. |
| GPT-4o | 0.80 | 0.86 | 0.68 | 0.79 | 0.77 | 0.59 |
| GPT-4o-mini | 0.78 | 0.77 | 0.62 | 0.75 | 0.66 | 0.54 |
| ChatGPT | 0.60 | 0.78 | 0.51 | 0.62 | 0.57 | 0.27 |
| Mistral-7B-Instruct-v0.2 | 0.64 | 0.68 | 0.55 | 0.50 | 0.43 | 0.43 |
| Phi-3-medium-4k-instruct | 0.73 | 0.76 | 0.66 | 0.61 | 0.61 | 0.63 |
| LLama-3-8B-Instruct | 0.65 | 0.72 | 0.55 | 0.57 | 0.53 | 0.45 |
| Gemma-2-9B-IT | 0.70 | 0.75 | 0.68 | 0.57 | 0.55 | 0.58 |
| Granite-3.0-8B-Instruct | 0.67 | 0.73 | 0.61 | 0.46 | 0.51 | 0.57 |
| Llama-3.1-8B-Instruct | 0.67 | 0.74 | 0.59 | 0.55 | 0.54 | 0.50 |
| LLama-3.2-1B-Instruct | 0.41 | 0.80 | 0.37 | 0.22 | 0.36 | 0.08 |
| LLama-3-70B-Instruct | 0.71 | 0.86 | 0.70 | 0.67 | 0.70 | 0.54 |
| Ministral-8B-Instruct-2410 | 0.63 | 0.81 | 0.55 | 0.66 | 0.67 | 0.31 |
| Phi-3.5-Mini-Instruct | 0.70 | 0.71 | 0.61 | 0.58 | 0.53 | 0.55 |
| Qwen-2.5-1.5B-Instruct | 0.51 | 0.76 | 0.45 | 0.48 | 0.55 | 0.19 |
| Qwen-2.5-14B-Instruct | 0.76 | 0.81 | 0.71 | 0.71 | 0.70 | 0.66 |
| Yi-1.5-34B | 0.67 | 0.78 | 0.65 | 0.62 | 0.56 | 0.48 |

Table 17: Count of examples in FOLIO(R; FOL) for each combination of (T)rue, (F)alse, and (U)ncertain label and predictions in that order. For example, TU is the number of times a LLM predicted a True label as Uncertain.

| Model | TT | TF | TU | FT | FF | FU | UT | UF | UU |
|---|---|---|---|---|---|---|---|---|---|
| GPT-4o | 1596 | 124 | 218 | 215 | 1004 | 224 | 329 | 359 | 932 |
| GPT-4o-mini | 1432 | 140 | 367 | 153 | 944 | 348 | 224 | 420 | 976 |
| ChatGPT | 1500 | 208 | 227 | 456 | 711 | 266 | 756 | 525 | 338 |
| Mistral-7B-Instruct-v0.2 | 1159 | 244 | 521 | 229 | 616 | 575 | 503 | 335 | 760 |
| Phi-3-medium-4k-instruct | 1405 | 120 | 393 | 278 | 688 | 457 | 365 | 226 | 1014 |
| LLama-3-8B-Instruct | 1471 | 193 | 249 | 462 | 621 | 332 | 674 | 396 | 532 |
| Gemma-2-9B-IT | 1378 | 112 | 408 | 256 | 659 | 489 | 397 | 268 | 919 |
| Granite-3.0-8B-Instruct | 1428 | 181 | 317 | 373 | 596 | 458 | 530 | 325 | 755 |
| Llama-3.1-8B-Instruct | 1449 | 163 | 305 | 358 | 647 | 409 | 575 | 289 | 725 |
| LLama-3.2-1B-Instruct | 1436 | 232 | 127 | 969 | 239 | 96 | 1142 | 237 | 117 |
| LLama-3-70B-Instruct | 1661 | 122 | 149 | 390 | 829 | 224 | 576 | 248 | 791 |
| Ministral-8B-Instruct-2410 | 1525 | 226 | 180 | 452 | 805 | 185 | 692 | 464 | 461 |
| Phi-3.5-Mini-Instruct | 1291 | 145 | 449 | 314 | 595 | 503 | 466 | 331 | 791 |
| Qwen-2.5-1.5B-Instruct | 1305 | 358 | 255 | 611 | 560 | 256 | 829 | 392 | 386 |
| Qwen-2.5-14B-Instruct | 1648 | 90 | 194 | 279 | 918 | 241 | 377 | 253 | 966 |
| Yi-1.5-34B | 1513 | 106 | 271 | 285 | 730 | 350 | 476 | 282 | 826 |

Table 18: Calculated precision and recall for each label in FOLIO(R; FOL).

| Model | True Label | | False Label | | Uncertain Label | |
|---|---|---|---|---|---|---|
| | Prec. | Rec. | Prec. | Rec. | Prec. | Rec. |
| GPT-4o | 0.75 | 0.82 | 0.68 | 0.70 | 0.68 | 0.58 |
| GPT-4o-mini | 0.79 | 0.74 | 0.63 | 0.65 | 0.58 | 0.60 |
| ChatGPT | 0.55 | 0.78 | 0.49 | 0.50 | 0.41 | 0.21 |
| Mistral-7B-Instruct-v0.2 | 0.61 | 0.60 | 0.52 | 0.43 | 0.41 | 0.48 |
| Phi-3-medium-4k-instruct | 0.69 | 0.73 | 0.67 | 0.48 | 0.54 | 0.63 |
| LLama-3-8B-Instruct | 0.56 | 0.77 | 0.51 | 0.44 | 0.48 | 0.33 |
| Gemma-2-9B-IT | 0.68 | 0.73 | 0.63 | 0.47 | 0.51 | 0.58 |
| Granite-3.0-8B-Instruct | 0.61 | 0.74 | 0.54 | 0.42 | 0.49 | 0.47 |
| Llama-3.1-8B-Instruct | 0.61 | 0.76 | 0.59 | 0.46 | 0.50 | 0.46 |
| LLama-3.2-1B-Instruct | 0.40 | 0.80 | 0.34 | 0.18 | 0.34 | 0.08 |
| LLama-3-70B-Instruct | 0.63 | 0.86 | 0.69 | 0.57 | 0.68 | 0.49 |
| Ministral-8B-Instruct-2410 | 0.57 | 0.79 | 0.54 | 0.56 | 0.56 | 0.29 |
| Phi-3.5-Mini-Instruct | 0.62 | 0.68 | 0.56 | 0.42 | 0.45 | 0.50 |
| Qwen-2.5-1.5B-Instruct | 0.48 | 0.68 | 0.43 | 0.39 | 0.43 | 0.24 |
| Qwen-2.5-14B-Instruct | 0.72 | 0.85 | 0.73 | 0.64 | 0.69 | 0.61 |
| Yi-1.5-34B | 0.67 | 0.80 | 0.65 | 0.53 | 0.57 | 0.52 |

Table 19: Number of examples of true positives (TP), false positives (FP), true negatives (TN), and false negatives (FN) for each LLM in LogiEval(R; PL). The counts for when the LLM was non-compliant with our prompt for positive (NP) and negative (NN) labels are also provided. Additionally, the calculated true positive rate (TPR), true negative rate (TNR), precision, and F1 score for each LLM is shown.

| Model | TP | FP | TN | FN | NP | NN | TPR | TNR | Prec. | F1 |
|---|---|---|---|---|---|---|---|---|---|---|
| GPT-4o | 1724 | 56 | 594 | 251 | 0 | 0 | 0.87 | 0.91 | 0.97 | 0.92 |
| GPT-4o-mini | 1310 | 116 | 534 | 665 | 0 | 0 | 0.66 | 0.82 | 0.92 | 0.77 |
| ChatGPT | 1348 | 260 | 390 | 625 | 2 | 0 | 0.68 | 0.60 | 0.84 | 0.75 |
| Mistral-7B-Instruct-v0.2 | 1240 | 187 | 379 | 640 | 95 | 84 | 0.66 | 0.67 | 0.87 | 0.75 |
| Phi-3-medium-4k-instruct | 1558 | 197 | 452 | 411 | 6 | 1 | 0.79 | 0.70 | 0.89 | 0.84 |
| LLama-3-8B-Instruct | 1246 | 209 | 434 | 715 | 14 | 7 | 0.64 | 0.67 | 0.86 | 0.73 |
| Gemma-2-9B-IT | 1464 | 217 | 432 | 497 | 14 | 1 | 0.75 | 0.67 | 0.87 | 0.80 |
| Granite-3.0-8B-Instruct | 1106 | 168 | 482 | 869 | 0 | 0 | 0.56 | 0.74 | 0.87 | 0.68 |
| Llama-3.1-8B-Instruct | 1523 | 242 | 400 | 430 | 22 | 8 | 0.78 | 0.62 | 0.86 | 0.82 |
| LLama-3.2-1B-Instruct | 985 | 250 | 342 | 820 | 170 | 58 | 0.55 | 0.58 | 0.80 | 0.65 |
| LLama-3-70B-Instruct | 1741 | 142 | 507 | 223 | 11 | 1 | 0.89 | 0.78 | 0.92 | 0.91 |
| Ministral-8B-Instruct-2410 | 1350 | 161 | 489 | 621 | 4 | 0 | 0.68 | 0.75 | 0.89 | 0.78 |
| Phi-3.5-Mini-Instruct | 1222 | 180 | 459 | 719 | 34 | 11 | 0.63 | 0.72 | 0.87 | 0.73 |
| Qwen-2.5-1.5B-Instruct | 1085 | 281 | 298 | 582 | 308 | 71 | 0.65 | 0.51 | 0.79 | 0.72 |
| Qwen-2.5-14B-Instruct | 1474 | 72 | 574 | 486 | 15 | 4 | 0.75 | 0.89 | 0.95 | 0.84 |
| Yi-1.5-34B | 1651 | 193 | 457 | 321 | 3 | 0 | 0.84 | 0.70 | 0.90 | 0.87 |

Table 20: Number of examples of true positives (TP), false positives (FP), true negatives (TN), and false negatives (FN) for each LLM in LogiEval(R; FOL). The counts for when the LLM was non-compliant with our prompt for positive (NP) and negative (NN) labels are also provided. Additionally, the calculated true positive rate (TPR), true negative rate (TNR), precision, and F1 score for each LLM is shown.

| Model | TP | FP | TN | FN | NP | NN | TPR | TNR | Prec. | F1 |
|---|---|---|---|---|---|---|---|---|---|---|
| GPT-4o | 1627 | 57 | 593 | 398 | 0 | 0 | 0.80 | 0.91 | 0.97 | 0.88 |
| GPT-4o-mini | 1084 | 95 | 555 | 941 | 0 | 0 | 0.54 | 0.85 | 0.92 | 0.68 |
| ChatGPT | 1346 | 177 | 472 | 678 | 1 | 1 | 0.67 | 0.73 | 0.88 | 0.76 |
| Mistral-7B-Instruct-v0.2 | 1172 | 159 | 403 | 716 | 137 | 88 | 0.62 | 0.72 | 0.88 | 0.73 |
| Phi-3-medium-4k-instruct | 1449 | 139 | 511 | 574 | 2 | 0 | 0.72 | 0.79 | 0.91 | 0.80 |
| LLama-3-8B-Instruct | 1267 | 139 | 502 | 752 | 6 | 9 | 0.63 | 0.78 | 0.90 | 0.74 |
| Gemma-2-9B-IT | 1355 | 118 | 532 | 665 | 5 | 0 | 0.67 | 0.82 | 0.92 | 0.78 |
| Granite-3.0-8B-Instruct | 1023 | 102 | 548 | 1002 | 0 | 0 | 0.51 | 0.84 | 0.91 | 0.65 |
| Llama-3.1-8B-Instruct | 1466 | 198 | 440 | 538 | 21 | 12 | 0.73 | 0.69 | 0.88 | 0.80 |
| LLama-3.2-1B-Instruct | 968 | 255 | 335 | 853 | 204 | 60 | 0.53 | 0.57 | 0.79 | 0.64 |
| LLama-3-70B-Instruct | 1630 | 116 | 533 | 392 | 3 | 1 | 0.81 | 0.82 | 0.93 | 0.87 |
| Ministral-8B-Instruct-2410 | 1304 | 161 | 486 | 708 | 13 | 3 | 0.65 | 0.75 | 0.89 | 0.75 |
| Phi-3.5-Mini-Instruct | 1059 | 140 | 493 | 922 | 44 | 17 | 0.53 | 0.78 | 0.88 | 0.67 |
| Qwen-2.5-1.5B-Instruct | 1013 | 220 | 374 | 775 | 237 | 56 | 0.57 | 0.63 | 0.82 | 0.67 |
| Qwen-2.5-14B-Instruct | 1303 | 89 | 555 | 707 | 15 | 6 | 0.65 | 0.86 | 0.94 | 0.77 |
| Yi-1.5-34B | 1489 | 156 | 494 | 534 | 2 | 0 | 0.74 | 0.76 | 0.91 | 0.81 |

