# OpenReview forum: "Autonomous Evaluation of LLMs for Truth Maintenance and Reasoning Tasks"
_ICLR.cc/2025/Conference — ICLR 2025 Poster_

### Official Review · Reviewer_t14Z · 2024-11-01

**Soundness:** 3
**Presentation:** 3
**Contribution:** 2
**Rating:** 5
**Confidence:** 5

**Summary:**

The paper introduces autoeval benchmark, designed for the autonomous evaluation of LLMs in formal tasks like truth maintenance and logical reasoning. The benchmark offers several features, such as dynamic dataset generation using CFGs and the use of formal verifiers to assess correctness. The authors argue that their approach mitigates common issues in existing benchmarks, like overfitting to static datasets, and provides a scalable method for evaluating LLMs without relying on human annotations. Empirical results show that performance on this benchmark correlates well with other established benchmarks.

**Strengths:**

- The use of CFGs to dynamically generate out-of-distribution datasets is a significant strength.
- The use of formal verifiers to check the semantic equivalence of LLM outputs is a good method that avoids brittle syntactic checks. This allows for a more robust evaluation of truth maintenance across different formal syntaxes.

**Weaknesses:**

- While the paper focuses on formal tasks like truth maintenance and logical reasoning, it remains unclear how well the proposed benchmark generalizes to less structured or more complex natural language tasks. The reliance on formal verifiers may limit its applicability to broader LLM use cases, such as creative writing or conversational AI, where formal correctness is less critical.
- Informalization, the process of converting formal syntax into natural language, is inherently ambiguous. The paper does not thoroughly address how it handles cases where multiple valid informalizations exist. This could lead to inconsistent evaluations when different LLMs produce semantically equivalent but syntactically different outputs.
- Weak evidence for the robustness of this method. While the paper claims that performance on autoeval correlates well with other benchmarks, the empirical validation seems somewhat limited. I would suggest that the authors test their method on a larger set of models. The current correlation results, while promising, could be influenced by noise or outliers, especially given the relatively small number of models tested, they might not be robust enough to generalize across different architectures or configurations. By increasing the number of models tested (e.g., to 50+), the authors could reduce the impact of potential outliers and noise in the data.

**Questions:**

- Given that formal verifiers are central to your evaluation process, how do you handle cases where the verifier fails or produces ambiguous results? Are there fallback mechanisms in place?
- Can your dynamic dataset generation approach be extended to more complex natural language tasks that do not rely on formal syntax? If so, how would this impact the scalability and performance of your evaluation framework?
- What steps have you taken to ensure that your benchmark remains relevant as LLMs evolve? Specifically, how do you plan to update the benchmark as new models with different architectures or capabilities are developed?
- In your empirical analysis, how many different LLMs were tested? It would be useful to know whether your results generalize across a broad range of models or are specific to a few.

---

> ### Author Response · Authors · 2024-11-18
> **Rebuttal (1/2)**
>
> Thank you for your review and detailed feedback. We are happy to address your concerns. We have also uploaded a new version of the paper for answering Q4 (Appendix L and onwards).
>
> ## $\S$ Regarding Question 4, _Generalization across a range of models_
> As shown in Figures 3, 5, and 6, we used a total of 7 different LLMs in our empirical evaluation. Specifically, we used the following models:
> * (OpenAI) GPT-3.5-turbo (gpt-3.5-turbo-0125)
> * (OpenAI) GPT-4o (gpt-4o-2024-08-06)
> * (OpenAI) GPT-4o-mini (o1-mini-2024-09-12)
> * (OpenAI) GPT-4o1-preview (o1-preview-2024-09-12, Out for less than 3 weeks before the paper deadline)
> * (Meta) LLama-3-8B-Instruct
> * (MistralAI) Mistral-v0.2-7B-Instruct
> * (Microsoft) Phi-3-medium-4k-instruct (14B)
>
> As mentioned in lines 344-345, these models provide a good mix of SOTA paid and open-source models, and thus, our results are an effective indicator of the current SOTA LLM technology. In addition, after reading your review, we started a few additional runs to provide a live estimate of the procses and illustrate the hands-free, time-efficient nature of our benchmarking platform. This was done by including a paid closed-source model (Anthropic Claude-3-Sonnet) and several open-source models which will elucidate in the next section. These runs were started on Nov-14-2024 15:00 UTC-12.
>
>
> We ran Claude for the ksat, propositional logic and regular expression dataset and the obtained results have been added in Appendix L.2 (Fig. 19). We cannot use Claude-3-Sonnet comprehensively for our experiments due to the exorbitant monetary cost required for running the full suite of our experiments in the paper. We will include these notes in the final version as well.
>
>
> ## $\S$ Regarding Weakness 3, _Weak evidence $\ldots$_
> As we mentioned above, our evaluation consists of 7 different models and provides a good mix of SOTA LLMs. Furthermore, our evaluation already matches when it doesn't exceed community established standards in terms of the numbers of LLMs used to illustrated and present research advances on metrics and benchmarks [1,2,3,4]. [1] compares with 9 models. [2] compares with 4. [3] compares with 8 and [4] compares with 5. These trends also hold amongst related work that we cite (e.g. (3) Tian et al, 2021; (4) Wu et al, 2022; (6) Saparov & He, 2023; (6) Patel et al, 2024; (3) Pan et al, 2024, (2) Jiang et al, 2024).
>
> Testing of 50 models is usually a collaborative effort for many benchmarks via platforms like HuggingFace (e.g. Big bench code, Open LLM leaderboard) and is not possible individually by us on our dataset due to the sheer magnitude of the costs involved in our experiments. We recognize the value of your comment and we have put 9 additional LLMs for evaluation through our benchmark.
>
> The 9 additional LLMs we are running are:
> | Model | Parameters | Link |
> |---|---|---|
> | Phi-3.5-mini-instruct | 4B | https://huggingface.co/microsoft/Phi-3.5-mini-instruct |
> | LLama-3.1-8B-Instruct | 8B | https://huggingface.co/meta-llama/Llama-3.1-8B-Instruct |
> | Llama3-8B-instruct-informalization | 8B | https://huggingface.co/Iddah/llama3-8B-instruct-informalization |
> | Ministral-8B-Instruct | 8B | https://huggingface.co/mistralai/Ministral-8B-Instruct-2410 |
> | Granite-8B-Instruct | 8B | https://huggingface.co/ibm-granite/granite-3.0-8b-instruct |
> | Gemma-2-9B-Instruct | 9B | https://huggingface.co/google/gemma-2-9b-it |
> | Qwen2.5-14B-Instruct | 14B | https://huggingface.co/Qwen/Qwen2.5-14B-Instruct |
> | Yi-34B-Chat | 34B | https://huggingface.co/01-ai/Yi-34B-Chat |
> | Llama-3-70B-instruct | 70B | https://huggingface.co/meta-llama/Meta-Llama-3-70B-Instruct |
>
> As mentioned above we included 9 additional LLMs based on your comments to illustrate the streamlined and automated nature of this process. We expect the results to complete in a few days (before the final deadline) and will update the paper and analyses when the data is available.
>
> [1] Microsoft.
> Phi-3 technical report.
> arXiv: https://arxiv.org/pdf/2404.14219
>
> [2]
> Zheng et al.
> Judging LLM-as-a-Judge with MT-Bench and Chatbot Arena
> NeurIPS-23
> https://openreview.net/forum?id=uccHPGDlao (Fig 3)
>
> [3]
> Zhuang et al
> ToolQA: A Dataset for LLM Question Answering with External Tools
> NeurIPS-23
> https://openreview.net/forum?id=pV1xV2RK6I
>
> [4]
> Gao et al
> LoRA: A Logical Reasoning Augmented Dataset for Visual Question Answering
> NeurIPS-23
> https://openreview.net/forum?id=bW1uwPV3im

---

> ### Author Response · Authors · 2024-11-18
> **Rebuttal (2/2)**
>
> ## $\S$ Regarding weakness 1
> As mentioned in lines 24-29, the focus of this paper has been to develop a benchmark for assessing LLM truth maintenance w.r.t. formal syntax. We use formal verifiers since accurately evaluating truth maintenance is paramount in applications involving formal syntax. Current SOTA benchmarks in this area cannot autonomously evaluate LLMs for truth maintenance and/or use datasets that do not scale. We agree that the use of formal verifiers restricts applicability to other LLM use cases such as creative writing but argue that benchmarks that evaluate creative writing similarly do not generalize to truth maintenance. The development of a unified benchmark for evaluating LLMs across a myriad of tasks is an interesting but difficult research problem that would be beyond the scope of our work.
>
> ## $\S$ Regarding Question 2
> As mentioned above, our focus is formal syntax-based tasks. We agree that extending our autonomous methodology to handle complex free-structured NL tasks like conversational AI is a very interesting direction for future work and is quite important as well. In such cases, the LLM would be expected to expand upon the truth or maintain truth that mattered. For example, prompting an LLM to write a creative story about a cat would require a benchmark that could evaluate whether the generated story is about a cat with other details disregarded. This is quite difficult since it is not clear what a general definition of equivalence is in such tasks. We hope that our platform can be one of the steps towards realizing such objectives.
>
> ## $\S$ Regarding weakness 2
>
> Our approach does handle (both through theory and experiments) the case of multiple informalizations. We show that our approach is robust against different informalizations of the same FS in lines 424-438, where we bound the probability of false positives that could occur from hallucinations.
>
> Firstly, in lines 107-112, we do acknowledge that LLMs can autoformalize or informalize differently. We provide an example of autoformalization in the paper but we will also include one of informalization to aid the clarity of the paper. We also note that the intuition that we require _equivalence_ and not _equality_ is one of the key strengths of our approach. Our definition of truth maintenance (lines 113-117) and our approach (lines 194-202) capture this aspect.
>
> As an example, let $\psi = p_1 \land p_1$ where $p_1 =$ _It is raining._
> Let $\psi' = p_1 \land p_1 = (\mathcal{A} \circ \mathcal{I})(\psi)$ be the result of our pipeline where $\varphi = \mathcal{I}(\psi) =$ _The sentence is a conjunction of two propositions, both of which specify that it is raining_ was the result of the informalization step. Similarly, let $\psi'' = (\mathcal{A} \circ \mathcal{I})(\psi) = p_1$  be the result of another invocation of our pipeline where $\varphi' = \mathcal{I}(\psi) =$ _Using the Idempotent Law, the sentence asserts that the weather is currently raining_ s.t. the informalizations were different $\varphi' \not= \varphi$. Our key intuition is that if $\psi \equiv \psi' \equiv \psi''$ then $\varphi \equiv \varphi'$.
>
>
> Finally, Appendix H showcases our standard deviation results, where we ran the same batch of the dataset 10 times. We checked the informalizations and they indeed are different for the same FS in a different run. Our results of this analysis (Fig. 8) show that there is not a significant variation in the results obtained in such a scenario empirically demonstrating the robustness of our pipeline.
>
>
> ## $\S$ Regarding Question 1
> Yes, there are fallback mechanisms in place. As mentioned in lines 524-525, we load verifiers with appropriate timeouts and logging mechanisms. Only 0.66% of all 85k entries of our results in Fig. 5 have encountered this and such errors are automatically excluded prior to the final outputs. We will improve the clarity of the main paper by including this.
>
> ## $\S$ Regarding Question 3
> A robust benchmark to mitigate benchmark data contamination was one of our key motivations behind this work (Sec 1). Thanks to the flexibility of our approach, we can dynamically generate new datasets by simply increasing the parse depth and/or changing the vocabularies. This allows our benchmark to generate OOD data as LLMs evolve. Furthermore, our code platform can be easily applied to new LLMs as evidenced by our showcasing of new experiments in our response to your concerns above. We also plan to take Reviewer caKYs suggestions of tools into account. Since the pipeline only requires FS as input and produces FS as output, new capabilities can be easily accomodated. For example, Tree-of-Thought as a technique can be easily added by simply calling an LLM that implements tree-of-thought.
>
> We hope that our responses have answered your questions and satisfied your concerns. We are happy to provide any additional details or clarifications that you need. We look forward to your response.

---

> ### Author Response · Authors · 2024-11-24
>
> Thank you for your response.
>
> We have uploaded a new version of the paper with more LLMs (16 in total) and updating the paper taking into considerations the changes you and the other reviewrs suggested. The changes are marked in teal color and new figures are framed in teal boxes.
>
> Specifically,
> * We added results with 10 new LLMs with diverse parameter sizes (from 1B to 70B) models. This represents a 166% increase from the 7 LLMs that we already had experimented on.
>     * The plots in the main paper have been updated to reflect these new results (Figures 3, 4, and 5)
>     * We have provided the complete models, their parameters, and the model versions in Table 7 in the Appendix.
>     * We have included all data  to compute the correlation and predictive power in Tables 7, 9 -- 14. We have also included Figures 18 and 19 with an easier view of all the models on our dataset.
>     * We have added Appendix N that refers to the new, comprehensive data.
>     * We have updated the overall analysis of the new results in lines 365-369.
> * We have also motivated the dynamic nature of our dataset by expanding on dataset diversity in Appendix N and lines 265-303.
> * We have provided more clarity that there can be different informalizations in lines 111-113.  We have also motivated this further in our theoretical analysis of bounding the false positive rate (lines 423-427)
> * We have made clear of the total number of examples that failed to verify due to a timeout in line 525.
> * We have reworded the text of our statement of Broader Impact (lines 514-519) to state that our approach can pave the way for developing automatic, human-free techniques for assessing truth maintenance of more free-form or creative tasks such as conversational AI.
> * We slightly reworded the conclusion to emphasize that our approach can easily be used to either generate new OOD datasets for evaluating new LLMs and be used as a surrogate benchmark.
>
> ### Key Takeaways
> Below are the key takeaways from our revised version
> * **Our overall results remain unchanged.** Our approach still retains a strong positive correlation and predictive power even with new LLMs introduced.
>     * For LogiEval(R; FOL), our results are now statistically significant. Previously, the results exhibited moderate correlation but were not statistically significant. The rest remain strongly positively correlated.
> * Our framework is highly flexible and can incorporate new (and evolving) LLMs with relative ease. We were not able to run few-shot results and LLM-as-a-verifier experiments on the new LLMs due to constraints on the compute resources available to us in this short time period. We ran a quick run on a small sample of our dataset on the experiments we could not run and we found that the trends still hold. We will include complete runs (few-shot and LLM-as-a-verifier) in the camera ready version.
> * Due to the increased number of permutations for predictive power due to the inclusion of new LLMs, the overall performance dropped 5%. We investigated the 12% drop in LogiEval(PL) and found that similar to FOL, GPT-4o-mini does not do well here. We have updated the analysis in lines 366-370 to reflect the same. Without this LLM, performance drop is 5% consistent with the observed drop across all other metrics.
> * Another characteristic that we found was that open-source models (esp the small ones like Llama-3-1b) are unusually verbose at times. This impacts the computations of the NL-based metrics like BLEU, ROUGE etc since the verbose sentences are compared against the hand-coded, terse human annotations from FOLIO($\mathcal{I}$). This further confirms our analysis in lines 397-410 of NL metrics not being good indicators of LLM performance for truth maintenance and showcases the strenghs of our approach.
>
> Please let us know if you have any further comments or concerns. We are happy to update the paper further.

---

> ### Author Response · Authors · 2024-11-27
> **Request for feedback**
>
> Dear Reviewer t14Z,
>
> As the period for uploading changes to the paper is ending soon, we would like to request that you please provide feedback on our revised version.
>
> Please let us know if our responses and new experiments have resolved your concerns with the paper.
>
> We have updated a new version and modified the comment above to reflect some of those new changes (partly because some of the figure numbers have changed).
>
> Specifically, we refer to our comment in the earlier repsonse.
>
> > We investigated the 12% drop in LogiEval(PL) and found that similar to FOL, GPT-4o-mini does not do well here. We have updated the analysis in lines 366-370 to reflect the same. Without this LLM, performance drop is 5% consistent with the observed drop across all other metrics.
>
>
> * We investigated the 12 percent performance loss in LogiEval and found out that LogiEval is a heavily imbalanced dataset where the positive class comprises 80 percent of all samples. Morever, this skew also persists when the problems are sorted based on difficulty with easier problems having a much larger proportion. We found that many models are more likely to just respond Yes and as a result accuracy is generally high. On the contrary, GPT-4o-mini etc actually tries to reason and thus makes mistakes sometimes. Thus, even thought GPT-4o-mini does better on our dataset, its performance is lower in LogiEval resulting in a loss of predictive power. We have updated the analysis in lines 366-370 to reflect the same. Taking into consideration this analysis, our autonomous approach retains its high predictive power and strong correlation with other, static, human-annotated LLM benchmarks.
>
> * We have updated the paper to reflect this analysis in lines 362-371. We have also added new Tables 15 -- 20 that provides the confusion matrices for the different models on these datasets. As we can see from Table 19, LogiEval is heavily skewed.
>
>
> We remain committed to improving our paper and would greatly appreciative your feedback.

---

> ### Author Response · Authors · 2024-12-02
>
> Hello Reviewer t14Z,
>
> As today is the last day for reviewers to engage in any discussion with the authors, we wanted to check whether you had any questions or feedback on our **revised version which adds 10 new LLMs to our experiments without impacting any of the results presented in the paper and addresses your other concerns as well.**
>
> We are happy to answer any questions that you might have or provide any clarifications.
>
> Respectfully,
>
> Authors of Submission 1397

---

### Official Review · Reviewer_JCXM · 2024-11-01

**Soundness:** 3
**Presentation:** 3
**Contribution:** 3
**Rating:** 6
**Confidence:** 3

**Summary:**

The ∀UTO∃∨∧L method evaluates the ability of LLMs to maintain truthfulness in formal tasks by dynamically generating datasets, utilizing the models' automatic formal and informal processing capabilities, and using formal verifiers. This approach allows for automated, human-free evaluation and can be extended to any formal grammar category that uses syntax and accepts equivalence checks. The authors demonstrate the effectiveness of ∀UTO∃∨∧L as an evaluation framework through empirical studies, revealing the limitations of current state-of-the-art models in maintaining truthfulness in formal tasks. By comparing it with existing benchmarks, the article highlights the practicality and importance of ∀UTO∃∨∧L in evaluating LLMs.

**Strengths:**

1. The article introduces the ∀UTO∃∨∧L benchmark as an innovative evaluation framework that can automatically assess the ability of large language models (LLMs) to maintain truthfulness in formal tasks. Its practicality lies in dynamically generating datasets, reducing the need for manual annotation, and providing an evaluation method without human intervention.
2. The ∀UTO∃∨∧L framework is designed to be flexible, allowing it to be extended to any formal grammar category that uses syntax and accepts equivalence checks.
3. The article demonstrates the correlation between the results of ∀UTO∃∨∧L and other existing benchmarks, indicating that ∀UTO∃∨∧L can serve as an effective alternative or complementary tool.

**Weaknesses:**

1. ∀UTO∃∨∧L relies on existing formal verifiers and parsing libraries to ensure evaluation accuracy. This means its reliability depends on the performance and accuracy of these tools, and any errors in them could affect the evaluation results of ∀UTO∃∨∧L.

2. While ∀UTO∃∨∧L reduces the need for manual annotation, generating dynamic datasets and performing formal verification may require significant computational resources, especially for large-scale datasets and complex logical structures.

**Questions:**

1. The article mentions using context-free grammars (CFGs) to generate datasets dynamically. How does this method ensure diversity and balance at different difficulty levels? Specifically in fields like logic and regular expressions, how does it ensure comprehensive coverage of various possible syntactic structures while maintaining a balanced sample size for each structure type?

2. The article evaluates whether LLMs can act as verifiers to assess the equivalence of formal grammar expressions and finds that even the most advanced LLMs struggle with this role. In the evaluation process, in which types of logical structures or specific conditions do LLMs perform the worst? Are there specific logical structures or syntactic features that make it difficult for LLMs to accurately judge equivalence?

---

> ### Author Response · Authors · 2024-11-18
> **Rebuttal (1/1)**
>
> We thank the reviewer for their review. We address your questions and concerns below. We have also uploaded a new copy of the paper with Appendix L onwards that we will refer to for answering Q2.
>
> ## $\S$ Regarding Weakness 1
> As noted in lines 535-537 in our submitted version, we agree that software bugs in some of the tools that we use are threats to validity of our approach. We argue that this is a common assumption (although not frequently mentioned) of _all_ works in the literature. Software bugs in any component can impact the output of the system for any approach for LLM-evaluation employed today.
>
> To mitigate such threats, rather than reimplementing our own verifiers from scratch, we have embraced open-source verifiers like Z3 (https://www.microsoft.com/en-us/research/project/z3-3/) and Prover9 (https://www.cs.unm.edu/~mccune/prover9/) which are quite popular with a robust community-based system for diagonising and fixing bugs, and have been widely used by many different branches of science . Similarly, we use NLTK, which is a widely used library for NL parsing. As a result, the overall threat to the validity of our platform due to bugs in such systems is quite low. We do motivate this in lines 297-301 from the lens of robust parsing but will note this positive aspect as well.
>
> $\S$ Regarding Weakness 2
> We agree that any external tool requires computational resources. However, the alternative for formal verifiers in this setting is to use LLM-based methods (e.g., LLM-as-a-judge) to provide a verification answer.  Our proposed approach of a CPU-based verifier is less compute-intensive than an LLM-based alternative like LLM-as-a-judge. This is because LLM forward inference requires several GPUs and thus quite a lot of compute to provide an answer. Furthermore, as our results (lines 418-422) show, LLMs cannot be used as verifiers in the truth maintenance task. Even strong LLMs with many billions of parameters like GPT-4o cannot provide accurate verification answers for modest-size FOL-S statements in one pass of forward inference, and smaller LLMs do worse.
>
> Another advantage of our use of popular libraries like Prover9 is that they are highly optimized and thus save computational costs further while providing a sound measure of accuracy.
>
> ## $\S$ Regarding Question 1
> Reviewer cakY also asked a similar question and we are glad to provide a clarification.
>
> Our dataset generation process is quite flexible and the user can use several hyperparameters (as detailed in Table 1) to influence the diversity of the datasets generated. For example, we used parse depth to generate expressions (or strings) at different levels of depth in the CFG parse tree. This is indicative of the descriptional complexity and is a well-known measure of difficulty (lines 155-157). Since CFGs can be used to generate sentences of unbounded length, we use a parameter that can allow fast generation of expressions  by pruning parts of the search space. Nevertheless, any string that the CFG can generate has a non-zero probability of being generated using our approach. If there is no pruning, then the probability of sampling an expression is simply the probability of sampling the given branch of the CFG parse tree.
>
> In our benchmark, our datasets can generate any propositional logic expression and all FOL expressions in the prenex normal form making it quite powerful for generating data.
>
> Different types of datasets can be easily generated by simply providing a different CFG. For example, the only difference between the K-SAT (Fig 2a) dataset and the Propositional Logic (Fig 2b) dataset is the CFG used in generating the formal expression. There were no code changes required and just a simple CFG to be written as input to the dataset generator.
>
> Finally, our Appendix contains metrics and metadata of the dataset (e.g. Fig 10), and as mentioned in the response to Reviewer caKY, we agree with both of you that this information could be improved and we will add the data from Appendix M in the final version.
>
> ## $\S$ Regarding Question 2
> Thank you for your question. We parsed the dataset and have provided an analysis in Appendix L.1, Fig. 18, Output Example 1, and Output Example 2 in the updated paper. We thank you for noting this since we believe this analysis improves our paper. We will update the main paper and appendix appropriately in the final version that we will upload. We will also update the code so that such plots can be generated for dynamically generated datasets.
>
> We hope our responses resolve your concerns and questions. We are happy to provide additional clarification if there are any further questions or comments.

---

> ### Author Response · Authors · 2024-11-24
>
> Thank you for your response.
>
> We have uploaded a new version of the paper with more LLMs (16 in total) and updating the paper taking into considerations the changes you and the other reviewers suggested. The changes are marked in teal color and new figures are framed in teal boxes.
>
> Specifically,
> * We added a discussion on dataset diversity in lines 265-298. We also made some clarity adjustments related to this in lines 300-303. We also added Appendix M that provides pointers to the various statistics on the diversity.
> * We expanded the text and description in lines 448-451 to provide some analysis of where LLMs struggle in verification. We also added Fig.18 and Appendix L that further discusses the results of using as LLMs as verifiers and also provide some analysis along with statistics of the data.
> * We have noted the advantage of our use of open-source, well-known tools like Prover9 in line 536 which makes threats to the validity of our results less.
>
> Please let us know if you have any further comments or concerns. We are happy to update the paper further.

---

> > ### Comment · Reviewer_JCXM · 2024-11-27
> > **Noted**
> >
> > Thanks for the authors' response. Noted, and I will keep my positive score.

---

> > > ### Author Response · Authors · 2024-11-27
> > >
> > > Thank you for your response.
> > >
> > > We uploaded a new version with some slight improvemens to the text (lines 1671-1720) that better discusses some of the errors LLMs made and made a minor tweak to the plots (Fig 20).
> > >
> > > Please do let us know if you have any further questions and concerns. We are happy to discuss.

---

### Official Review · Reviewer_cakY · 2024-11-04

**Soundness:** 4
**Presentation:** 4
**Contribution:** 4
**Rating:** 8
**Confidence:** 4

**Summary:**

The paper presents a new approach to evaluate and benchmark LLMs on formal syntax. The authors suggest using CFGs to generate synthetic data for testing the LLMs, they utilize formal verifiers to verify the logic.

Contributions:
1. A new method to evaluate LLMs on formal syntax without using static datasets by generating synthetic data using CFGs and adding vocabulary using another LLM
2. Their method has better indication of accuracy than other metrics for the LLM performance on various FOL.

**Strengths:**

1. LLMs can be now tested on dynamic data generated by this framework for Formal Syntax
2. Their method has better indication of accuracy than other metrics for the LLM performance on various FOL.
3. Very good experimentations , dataset size and presentation

**Weaknesses:**

1. Explanation on how the generated dataset is having diverse data points or quality is missing.

Update: Rebuttal answered various points about this and mentioned they have added more information in appendix.

**Questions:**

1. How diverse can the generated dataset be ?
2. How do you ensure that duplicate data points won’t be there ? I.e data points with very close similarity or minute changes or only change in vocabulary while the logic statement is same
3. Would love to know how LLMs with tools work as verfiers

---

> ### Author Response · Authors · 2024-11-18
> **Rebuttal (1/1)**
>
> Thank you for your review. We address your concerns and questions below. We have also added an updated section in the paper (Appendix L onwards) that indicates new changes to address the reviews. We will consolidate these to improve the text of the main paper and appendix in the final version.
>
> ## $\S$ Weakness 1 and Questions 1, 2
> > Explanation on how the generated dataset is having diverse data points or quality is missing.
>
> The dataset generation process was discussed in detail in Appendix B and details on diversity and quality were mentioned in lines 823-831 of the submitted paper. Our dataset generation process guarantees that any expression that can be generated at a given depth in the CFG is sampled with a non-zero probability.
>
> We also provide some measure of the dataset diversity by reporting the dataset statistics. For example, in Fig. 10 (Appendix), we provide some metrics on the # of examples with a given number of operators for logic-based formal syntax. Row 1, column 2 in that figure illustrates that for Propositional Logic, there are roughly an equal number of examples with $n \in [0-12]$ $\land$ operators in the expression. We also made a general comment on the diversity in lines 1285-1295 when analyzing LLM performance alongside metrics other than the CFG parse tree depth.
>
> Based on the answers we provide to your questions 1 and 2 below, we will reword Appendix B in the final version that we will upload.
>
> > How diverse can the generated dataset be?
>
> As mentioned above, any string in the CFG has a non-zero probability of getting sampled. Thus, we can generate any expression from the language that the CFG represents. We can determine the probability of each expression getting sampled by simply computing the probabilities by following the branches of the parse tree (each branch is sampled uniformly).
>
> > How do you ensure that duplicate data points won't be there ? I.e data points with very close similarity or minute changes or only change in vocabulary while the logic statement is same.
>
> As mentioned in line 246, the total number of unique samples in our dataset is around 85k (170k with few-shot prompts). Our definition of uniqueness counts the number of samples s.t. $FS_i = FS_j$ for any $i,j$. We clean each FS to remove unnecessary parentheses etc.
>
> > I.e data points with very close similarity or minute changes or only change in vocabulary while the logic statement is same.
>
> This is a great point. Both $p_1 \land p_2$ and $p_2 \land p_1$ may be present in our dataset. We argue that our definition of uniqueness makes our dataset robust since it guards against positional bias and other types of biases that LLMs might be susceptible to. For the example above, one would expect an LLM to perform the same in both expressions since they are equivalent. Failure to maintain truth on $p_2 \land p_1$ but not on $p_1 \land p_2$ might reveal some bias in the LLM.
>
> Regarding similarity w.r.t. vocabulary:
> For a given formula $p_1 \land p_2$, Alg. 1 can replace $p_1$ and $p_2$ with different elements from the vocabulary set. These are useful since the vocabulary can often influence the LLM's informalization, even if the underlying logic statement is the same.
> For example, for a formula $p_1 \lor p_2$ where $f_1 = \textit{Human}(x) \land \textit{Mortal}(x)$ and $f_2 = \textit{WorksOut}(x, y, z) \land \textit{Profession}(x, a)$ allows testing of different arities of predicates.
>
> Our dataset does contain such occurrences, but the probability of such instances decreases as the parse depth increases since there is a lower chance of equivalent expressions getting sampled.  We have updated the rebuttal (Appendix M) with this data and will include it in the final revision as well. To summarize, for propositional logic: the total unique samples by replacing all propositions $p_i$ in a formula $f$ with a single proposition $p$ (e.g. $p_1 \land p_2 \ldots) \rightarrow (p \land p \ldots)$ is 17.7k out of a dataset size of 19.6k (90%). Similarly, for k-sat this is 85%.
>
> Finally, our dataset generator provides configuration parameters to control the diversity and the number of samples with different vocabularies for the same logic expression. A user can use these parameters to diversify (or undiversify) the dataset (by increasing the sample count on the branching factor etc). Table 1 lists the hyperparameters that we used for the results reported in the paper.
>
> ## $\S$ Regarding question 3
> We really like your suggestion. Our analysis of _Can LLMs serve as verifiers_ in lines 439-444 only provided two formulae to the LLM and asked it to (either via chain-of-thought or directly) answer whether they were equivalent or not. We have not explored the use of tool calling. This is a great suggestion and we will include this as future work to enhance our benchmark.
>
> We hope our responses resolve your concerns and questions. We are happy to discuss further. Please let us know if you have any additional questions.

---

> > ### Comment · Reviewer_cakY · 2024-11-22
> >
> > Thank you very much for the rebuttal.
> >
> > > This is a great point. Both \( p_1 \land p_2 \) and \( p_2 \land p_1 \) may be present in our dataset. We argue that our definition of uniqueness makes our dataset robust since it guards against positional bias and other types of biases that LLMs might be susceptible to. For the example above, one would expect an LLM to perform the same in both expressions since they are equivalent. Failure to maintain truth on \( p_2 \land p_1 \) but not on \( p_1 \land p_2 \) might reveal some bias in the LLM.
> >
> > This is a great point, I suggest adding this in the main paper.
> >
> > > We really like your suggestion. Our analysis of Can LLMs serve as verifiers in lines 439-444 only provided two formulae to the LLM and asked it to (either via chain-of-thought or directly) answer whether they were equivalent or not. We have not explored the use of tool calling. This is a great suggestion and we will include this as future work to enhance our benchmark.
> >
> > Please mention about this and any other future directions for this work.
> >
> > I am happy with all the answers and I believe this is a very good work. I am more confident of my evaluation and hence I am increasing my confidence and contribution score.

---

> ### Author Response · Authors · 2024-11-24
>
> Thank you for your response.
>
> We have uploaded a new version of the paper with more LLMs (16 in total) and updating the paper taking into considerations the changes you and the other reviewers suggested. The changes are marked in teal color and new figures are framed in teal boxes.
>
> Specifically,
> * We added a discussion on dataset diversity in lines 265-298. We also made some clarity adjustments related to this in lines 300-303.
> * We also made a note of your suggestion to include the use of tools as potential future work in lines 536-537.
>
> Please let us know if you have any further comments or concerns. We are happy to update the paper further.

---

### Meta-Review · Area_Chair_NHKH · 2024-12-21

**Metareview:**

This paper presents $\forall$uto$\exists$$\lor\land$L, an innovative autonomous benchmark for evaluating LLMs on formal reasoning tasks. The key strengths include: (1) dynamic dataset generation using CFGs that prevents overfitting, (2) automated ground truth generation eliminating human annotation needs, and (3) formal verifiers ensuring semantic correctness. The main findings demonstrate strong correlation between performance on $\forall$uto$\exists$$\lor\land$L and other established benchmarks. While initially tested on 7 LLMs, the authors expanded to 16 diverse models (4B-70B parameters) during rebuttal, strengthening the empirical validation. The paper makes a clear contribution by introducing a scalable, automated evaluation framework for formal reasoning capabilities in LLMs.

**Additional Comments On Reviewer Discussion:**

The discussion focused on three main concerns: limited model coverage, dataset diversity, and generalizability. The authors thoroughly addressed these by: (1) expanding from 7 to 16 LLMs spanning 4B-70B parameters, with results maintaining strong correlations; (2) providing detailed analysis of dataset generation and diversity metrics in Appendix M; and (3) clarifying the framework's scope for formal reasoning while acknowledging limitations for creative tasks. Reviewer cakY (8/10) and JCXM (6/10) were satisfied with the responses, while t14Z's (5/10) concerns about model coverage were directly addressed through comprehensive expansion. The revisions strengthened the paper's empirical validation without changing its core findings, supporting acceptance.

---

### Decision · Program_Chairs · 2025-01-22

Accept (Poster)